# Mechano-responsive hydrogen-bonding array of thermoplastic polyurethane elastomer captures both strength and self-healing

Youngho Eom[1,5], Seon-Mi Kim[2,5], Minkyung Lee[2], Hyeonyeol Jeon [2], Jaeduk Park[3], Eun Seong Lee[3], Sung Yeon Hwang [2,4✉], Jeyoung Park [2,4✉] & Dongyeop X. Oh [2,4✉]

Self-repairable materials strive to emulate curable and resilient biological tissue; however, their performance is currently insufficient for commercialization purposes because mending and toughening are mutually exclusive. Herein, we report a carbonate-type thermoplastic polyurethane elastomer that self-heals at 35 °C and exhibits a tensile strength of 43 MPa; this elastomer is as strong as the soles used in footwear. Distinctively, it has abundant carbonyl groups in soft-segments and is fully amorphous with negligible phase separation due to poor hard-segment stacking. It operates in dual mechano-responsive mode through a reversible disorder-to-order transition of its hydrogen-bonding array; it heals when static and toughens when dynamic. In static mode, non-crystalline hard segments promote the dynamic exchange of disordered carbonyl hydrogen-bonds for self-healing. The amorphous phase forms stiff crystals when stretched through a transition that orders inter-chain hydrogen bonding. The phase and strain fully return to the pre-stressed state after release to repeat the healing process.

[1] Department of Polymer Engineering, Pukyong National University, Busan 48513, Republic of Korea. [2] Research Center for Bio-based Chemistry, Korea Research Institute of Chemical Technology (KRICT), Ulsan 44429, Republic of Korea. [3] Department of Biomedical Chemical Engineering, The Catholic University of Korea (CUK), Bucheon-si 14662 Gyeonggi-do, Republic of Korea. [4] Advanced Materials & Chemical Engineering, University of Science and Technology (UST), Ulsan 44429, Republic of Korea. [5]These authors contributed equally: Youngho Eom, Seon-Mi Kim ✉email: crew75@krict.re.kr; jypark@krict.re.kr; dongyeop@krict.re.kr

Self-healing materials that autonomously and repeatedly repair physical damage through dynamic physical or chemical bonds[1–7] have gained significant levels of attention in a variety of material fields, including automobile components[8,9], electronics[10–14], robotics[15,16], and healthcare devices[17–20]. However, the trade-off between mechanical and healing performance is the biggest hurdle that these materials face and need to overcome. The high chain rigidity, entanglement, and crystallinity required for mechanical strength conflict with the high diffusibility and exchange of dynamic bonds required for repairing damage[21–24]. Consequently, materials that autonomously heal at ambient temperature exhibit mechanical performance that is unsatisfactory for industrial commercialization.

In contrast, biological tissue is not only able to efficiently self-repair but is also very tough and resilient[25,26]. The secrets of some biological mechano-responsive processes that toughen biomaterials have recently been unveiled, but such insight remains undiscovered or undeveloped for man-made self-healing materials[27,28]. For example, the egg-capsule wall of a channeled whelk effectively absorbs shock with a high degree of reversible extensibility and stiffness[29]; its constituent proteins undergo reversible structural α-helix to β-sheet transitions with internal-energy changes when loaded. After the removal of stress, energy relaxation results in the recovery of a high degree of strain. As another example, the toughening and shape memory of spider silk is the result of the arrangement and density of the intermolecular hydrogen (H)-bonds in silk crystals that are tuned by the spinning rate from the spider's secretion gland[30–32].

Herein, we report a self-healing carbonate-type thermoplastic polyurethane (TPU) elastomer with remarkable ultimate tensile strength (UTS) of 43 MPa for a self-healing elastomer that functions at ambient temperature; this TPU even outperforms a commercial TPU used as a footwear component[33–38]. The rich carbonyl groups of the carbonate soft segments form H-bonds. As opposed to a typical TPU, its hard segments are poorly stacked, thereby providing a fully amorphous phase as well as poor phase separation. In addition, this TPU exhibits favorable biocompatibility. Thus, this material holds great potential as a structural material for electronic skin (e-skin) and soft robotics in healthcare systems.

Mechano-responsive H-bond biomimicry involving the unique characteristics of the respective hard and softs segments is the key to the great performance of this material. The H-bonding array of carbonyl groups undergoes a reversible structural change upon external stress. In static mode, the amorphous phase enables the segmental swapping of H-bonds and the dynamic formation of aromatic disulfides, which enables self-repair in the absence of stress, with molecular flow retarded due to the abundant carbonyl H-bonds that increase the elastic modulus. Under strain, the amorphous phase is transformed into a rigid metastable crystal where ordered H-bonds form between stretched chains to endure heavy loads. After removal of the stress, the metastable crystal returns to the amorphous phase, with its shape and strain fully recovered, and this elastomer is immediately available for self-healing.

## Results
### Preparing a strong self-healing elastomer that functions at ambient temperature.
The self-healing TPU referred to as C-IP-SS was synthesized from poly(hexamethylene carbonate) diol (C, a carbonate-type aliphatic macrodiol) as the soft segment, asymmetric alicyclic isophorone diisocyanate (IP) as the hard segment, and an aromatic disulfide (SS) as the chain extender (Fig. 1a and Supplementary Fig. 1)[39–44]. SS metathesis drives intrinsic self-healing at room temperature[45,46]. Our previous self-healing TPU (referred to as E-IP-SS), which contains ether-type macrodiol (E), IP, and SS units[47], was used as the control. The number-average molecular weights ($M_n$s) of the carbonate- and ether-type macrodiols are 1 kg mol$^{-1}$ and their compositions in each self-healing TPU are almost equal (15 wt%) (Supplementary Figs. 2-6 and Supplementary Table 1). Commercial TPU (Es-MD), containing ester-type macrodiol (Es) and methylene diphenyl diisocyanate (MD) units, was used as a non-self-healable control. The weight-average molecular weights ($M_w$s) of C-IP-SS, E-IP-SS, and Es-MD were 24, 46, and 99 kg mol$^{-1}$, respectively.

Interestingly, C-IP-SS exhibited significantly superior mechanical properties compared to other self-healing materials. The tensile properties of the three types of TPU film prepared by solvent casting were examined. Prior to this study, E-IP-SS exhibited the highest toughness (26.9 MJ m$^{-3}$) and UTS (6.8 MPa) among self-healing materials that function at ambient temperature (Fig. 1b). Nevertheless, its tensile performance is trivial compared to that of Es-MD, with toughness and UTS of 115 MJ m$^{-3}$ and 36 MPa, respectively. Although C-IP-SS has the lowest $M_w$ among the TPUs, its UTS of 43 MPa is higher than the other two control TPUs. It exhibits two-fold lower extensibility (450%) but a 2.8-fold greater toughness (75 MJ m$^{-3}$) than E-IP-SS (Fig. 1a, c).

The outstanding tensile properties of C-IP-SS have little penalty on its self-healing efficiency. C-IP-SS films were cut in half, reattached, and then allowed to recover at 35 °C for varying amounts of time (Fig. 1b, c and Supplementary Table 2). UTS recoveries of 29%, 39%, 65%, and 77% were observed after 1, 6, 24, and 48 h, respectively. Notably, the UTS of the recovered C-IP-SS exceeded that of virgin E-IP-SS after only 1 h of self-healing; after 48 h, the recovered C-IP-SS is as strong as Es-MD (Fig. 1c, d and Supplementary Movie 1), and another cut and re-spliced film with a contact cut-area of 30 mm × 3 mm was able to readily lift a 10-kg weight (Fig. 1e and Supplementary Movie 2). Even after 1 min of recovery at room temperature (25 °C), a cut and re-spliced film with a contacted cut-area of 5 mm × 1 mm was able to withstand manual drawing and twisting (Fig. 1f and Supplementary Movie 3). According to the Ashby plot of UTS versus self-healing temperature for recently reported elastomers, previous tensile strengths did not exceed 20 MPa without the aid of light or a temperature above 40 °C (Fig. 2 and Supplementary Table 3)[9,21–24,34–40,42,43,47–51].

### Achieving both self-healing and strength through reversible structural change.
The breakthrough in the trade-off between self-healing and strength achieved for C-IP-SS is attributable to a dual mode of operation involving its mechano-responsive H-bonding array, with reversible strain-induced crystallization providing a major contribution. In the stress–strain curve for C-IP-SS (Fig. 1b), the concave upward trend is a clear sign of strain hardening. Figure 3a reveals that the tensile moduli (i.e., differential stress values) increase sharply with strain and are substantially different from those of E-IP-SS and Es-MD. C-IP-SS exhibits a more than 4.3-fold higher modulus than the other TPUs at 400% elongation, and becomes translucent due to the formation of crystallites, while the other TPUs remain relatively transparent (Fig. 3b, c and Supplementary Figs. 7 and 8). Interestingly, the original strain and transparency of C-IP-SS are fully recovered upon unloading (Supplementary Fig. 9 and Supplementary Movie 4); strain-induced crystallization upon loading is superficially incompatible with full recovery upon unloading; it cannot be explained either by the solid crystallization of semi-crystalline polymers or entropic elastomer elasticity because the former does not facilitate strain recovery, while the latter does not enable phase transition. The strain-induced phase transition

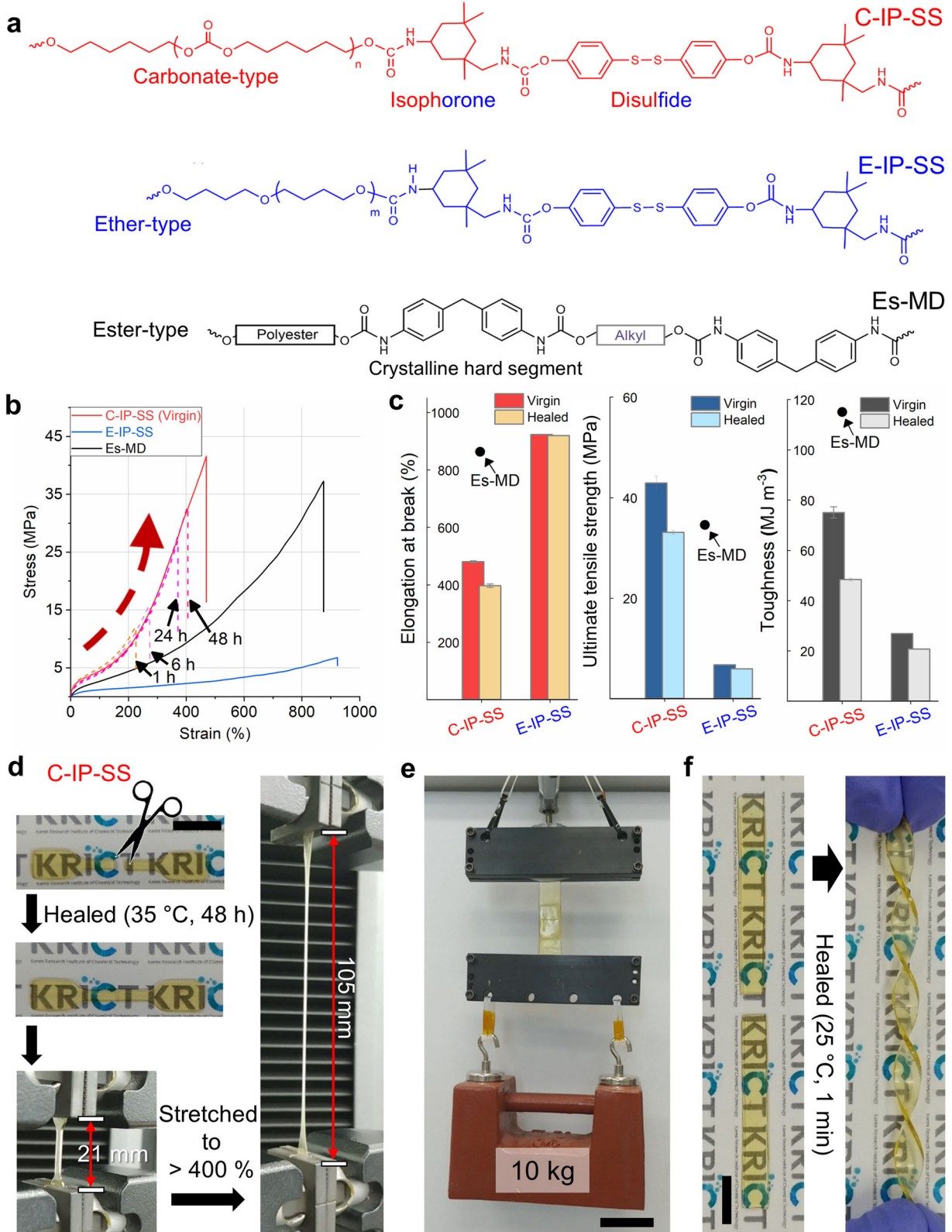

**Fig. 1 Chemical structures and mechanical properties of self-healable TPUs. a** Chemical structures and **b** tensile stress–strain curves for the three TPUs: C-IP-SS, E-IP-SS, and commercial Es-MD. Tensile curves of cut and rejoined C-IP-SS with different healing times at 35 °C are included. **c** Comparing mechanical properties, such as elongation at break, ultimate tensile strength, and toughness, for virgin and healed TPUs. The value for the non-self-healable Es-MD is given as a point in each panel. **d** Photographic images of cut and healed C-IP-SS films before (left) and after stretching up to 400% (right) (scale bar: 1 cm). **e** C-IP-SS film (thickness: 0.3 cm) cut in half, rejoined, and healed for 48 h at 35 °C, followed by a 10-kg weight-lifting demonstration (scale bar: 5 cm). **f** Photographic images of a twisted C-IP-SS film (thickness: 0.1 cm) after healing at 25 °C for 1 min (scale bar: 1 cm).

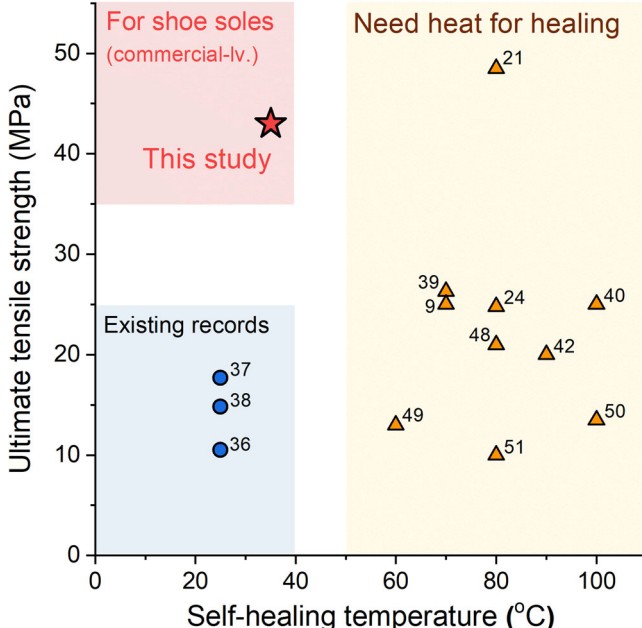

**Fig. 2 Ashby plot of ultimate tensile strength versus self-healing temperature of C-IP-SS and other elastomers reported in literature.** The red star indicates the properties of C-IP-SS. The blue circles indicate the properties of the reported autonomously self-healing elastomers without external stimulus. The yellow triangles indicate the properties of the reported elastomers that need heat for healing.

($\alpha$-helix $\leftrightarrow$ $\beta$-sheet) in some natural elastomeric materials is involved in their reversible deformation upon stretching, which results in long-range elastic recovery[52]. In other words, the internal energy change through this phase transition is a primary driving force for elasticity as opposed to the entropic behavior of a conventional rubber or elastomer. Inspired by the recovery process of biomaterials, we deduce that rapid switching between the two individual internal structures drives the reversible deformation of C-IP-SS (Fig. 3d).

The unusual strain-induced crystallization of C-IP-SS is strongly correlated with the reversible disorder-to-order switching of the mechano-responsive H-bonding array. Therefore, we separately investigated the internal H-bonding structures of C-IP-SS in its static and dynamic states by Fourier-transform infrared (FT-IR) spectroscopy (Fig. 4). As evidenced by differential scanning calorimetry (DSC), C-IP-SS is poorly microphase separated, with non-crystalline hard segments in the static state because the non-planar SS and IP moieties interfere with hard-segment stacking (Supplementary Fig. 10)[53]. This behavior is unusual because typical TPUs have distinctly separated amorphous soft-domain and crystalline hard-domain microphases induced by the intermolecular stacking of mesogenic MD. The FT-IR spectrum of Es-MD shows an intense band at 1703 cm$^{-1}$ that corresponds to H-bonded urethane groups in the hard domain (Supplementary Fig. 11), and conventional carbonate-type TPUs that contain MD also exhibit two dominant bands at approximately 1736 and 1699 cm$^{-1}$ from their respective soft and hard domains[41,54]. On the contrary, the FT-IR spectrum of C-IP-SS shows four bands associated with carbonyl (C = O) groups (Fig. 4a); free (non-H-bonded) carbonyl groups of hard and soft

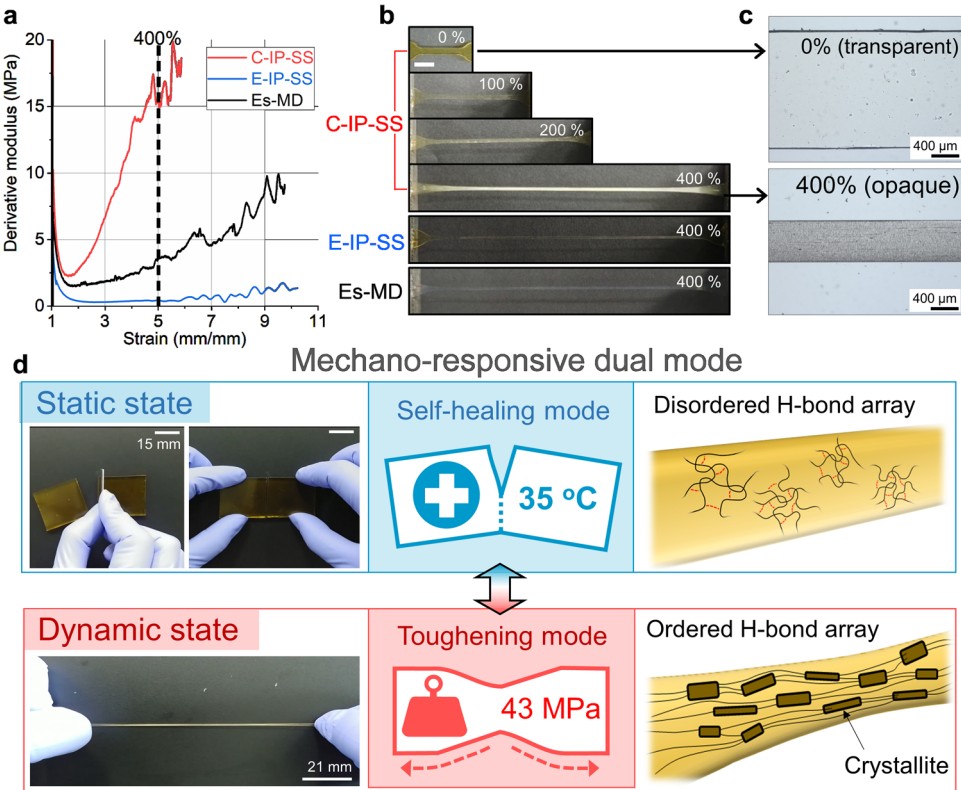

**Fig. 3 Mechano-responsive dual-mode of C-IP-SS. a** Variations in tensile modulus versus strain obtained from the first derivatives of the tensile stress–strain curves of the TPUs (Fig. 1b). **b** Macroscopic structural changes undergone by the TPUs upon uniaxial stretching to 400% (scale bar: 6 mm). **c** Optical microscopy images of unstretched and 400%-stretched C-IP-SS specimens. **d** Schematic illustration of mechano-responsive dual-mode operation through the reversible disorder-to-order transition of the H-bond array: self-healing in the static state and toughening in the dynamic state.

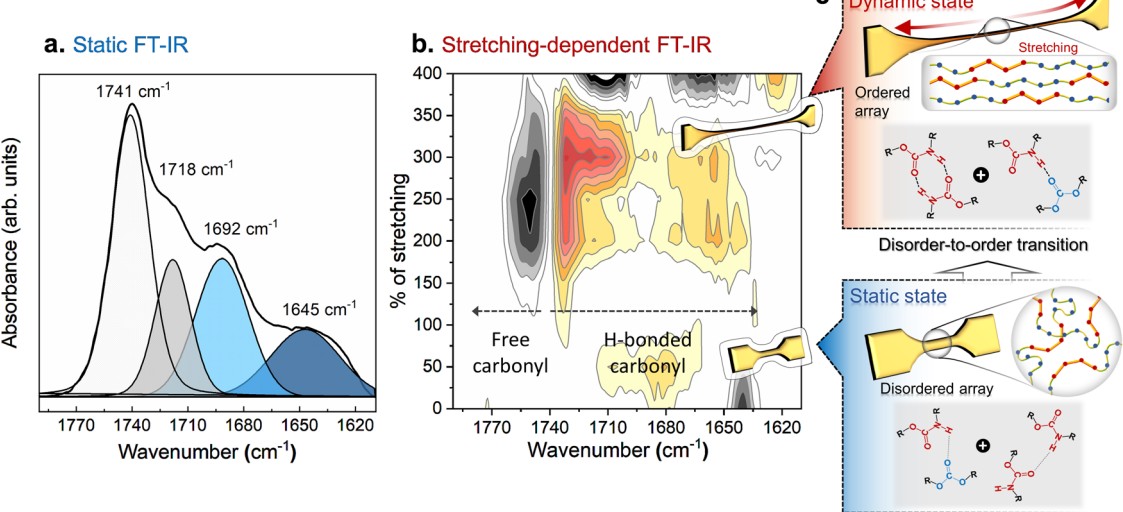

**Fig. 4 FT-IR analysis of C-IP-SS in the static and dynamic states. a** The carbonyl region of the FT-IR spectrum of C-IP-SS with peak deconvolution. **b** 2D gradient FT-IR map with respect to stretching percentage. The red contour lines represent positive values of d$A$/d$E$ (the first derivative of the absorbance ($A$) as a function of the extension % ($E$)); i.e., an increasing trend in absorbance as a function of the degree of extension, and vice versa, for the gray contour lines. **c** Schematic of the mechano-responsive changes in chain alignment and the associated H-bond arrays for C-IP-SS.

segments (1741 cm$^{-1}$), H-bonded carbonyl groups between hard and soft segments (1718 cm$^{-1}$), and H-bonded carbonyl groups between hard segments (1692 cm$^{-1}$)[55,56]. The shoulder peak at a lower frequency (1645 cm$^{-1}$) corresponds to sterically hindered ester carbonyl groups that neighbor SS units[57]. As a result, the band at 1718 cm$^{-1}$ due to H-bonding between hard and soft segments is as strong as the band at 1692 cm$^{-1}$ associated with H-bonding between hard segments. The –NH– units of each hard segment can form H-bonds with abundant soft segmental carbonyl groups[58]. Therefore, in static mode, C-IP-SS has a relatively high degree of disorder with regard to its non-crystalline hard segments and the high concentration of H-bond-free carbonyl groups.

Uniaxial stretching in the dynamic state enhances the degree of H-bond ordering, leading to an increase in crystallinity and a decrease in the concentration of carbonyl groups that are not H-bonded, which is well-represented by a 2D gradient FT-IR map, in which the first derivative of the absorbance ($A$) as a function of the extension % ($E$) (i.e., d$A$/d$E$) is constructed in $E$ versus wavenumber space (Fig. 4b)[59]. The red and gray contours indicate positive and negative d$A$/d$E$, respectively. The strongest red and gray contour lines at 1718 and 1741 cm$^{-1}$, respectively, confirm that extension strengthens the adsorption band corresponding to H-bonding between hard and soft segments, but weakens the band associated with the absence of H-bonding or disordered soft segments, which are ascribable to the disorder-to-order transition of H-bonds between the laterally aligned stretched carbonate chains (i.e., crystallites) (Fig. 4c and Supplementary Fig. 12). As expected, the absorption-band pattern fully returns to that of the original state after release. Thus, the mechano-responsive H-bonding structure undergoes array switching.

Rheology was used to explore whether or not the internal H-bonded structure affects the mechanics of C-IP-SS in static mode. Oscillating rheological testing at very low strain (1%) is a useful tool for investigating the static mode because it measures linear viscoelastic behavior. Each TPU was subjected to frequency sweep testing at 25 °C (Fig. 5a and Supplementary Fig. 13) to provide its complex viscosity at 0.05 rad s$^{-1}$ ($\eta^*_{0.05}$), yield stress, and the slope of the Cole−Cole plot. In short, higher values of $\eta^*_{0.05}$ and yield stress, and lower Cole–Cole-plot slopes indicate that C-IP-SS is a more solid-like system[20,60].

The temperature-dependent loss tangents (tan $\delta$) for the three TPUs were investigated in the 25–95 °C range (Fig. 5b). If tan $\delta$ shows a decrease (or increase) by a standard value of unity, then the system exhibits solid-like (or liquid-like) behavior[61]. In contrast to the tan $\delta$ value of E-IP-SS, that of C-IP-SS remained below unity over the entire examined temperature range and was lowest near room temperature, which is a meaningful result because it reveals that a product made of C-IP-SS will be relatively safe from melting at high temperature. Moreover, C-IP-SS exhibited the highest Young's modulus (elastic modulus at a low tensile strain) of 15.5 MPa, which is 11-, and 1.7-fold higher than those of E-IP-SS, and Es-MD, respectively (Supplementary Table 2). These data suggest that C-IP-SS is the most solid-like TPU among those studies owing to its rich carbonyl H-bonds, even though it has a four-fold lower $M_w$ than Es-MD. As revealed by FT-IR spectroscopy, H-bonds are created between the hard segments of C-IP-SS and all other hard and soft segments to balance the disordered chains, while the H-bonds in Es-MD are relatively fixed between ordered hard segments, but are lacking between soft segments or disordered chains. Consequently, the former system exhibits a high degree of disorder while constructing a structure with a higher density of internal H-bonds than the latter.

As a motif for self-healing, the molecular process in static mode can be described by rheological master curves of storage ($G'$) and loss ($G''$) moduli (Fig. 5c). A typical viscoelastic polymer shows four zones that are characterized by trends in $G'$ and $G''$ (Fig. 5d) that are ordered as terminal, rubbery plateau, transition, and glassy zones[62]. Comparing the two self-healable TPUs reveals that E-IP-SS only exhibits the first two zones, with a clear terminal region ($G'' > G'$). On the other hand, C-IP-SS shows three zones, from terminal to transition, with a dominant rubbery plateau region ($G' > G''$). These observations suggest that the chains in C-IP-SS experience a lower degree of molecular slips than those in E-IP-SS. As a quantitative parameter, the flow transition relaxation time ($\tau_f$) was determined from the reciprocal frequency at the $G'/G''$ crossover between the terminal and rubbery zones, which was found to be >2 years for C-IP-SS and 112 s for E-IP-SS (Fig. 5e), suggesting that C-IP-SS behaves more like a glassy solid than E-IP-SS; hence, chain-flow relaxation barely contributes to C-IP-SS healing on a reasonable time scale.

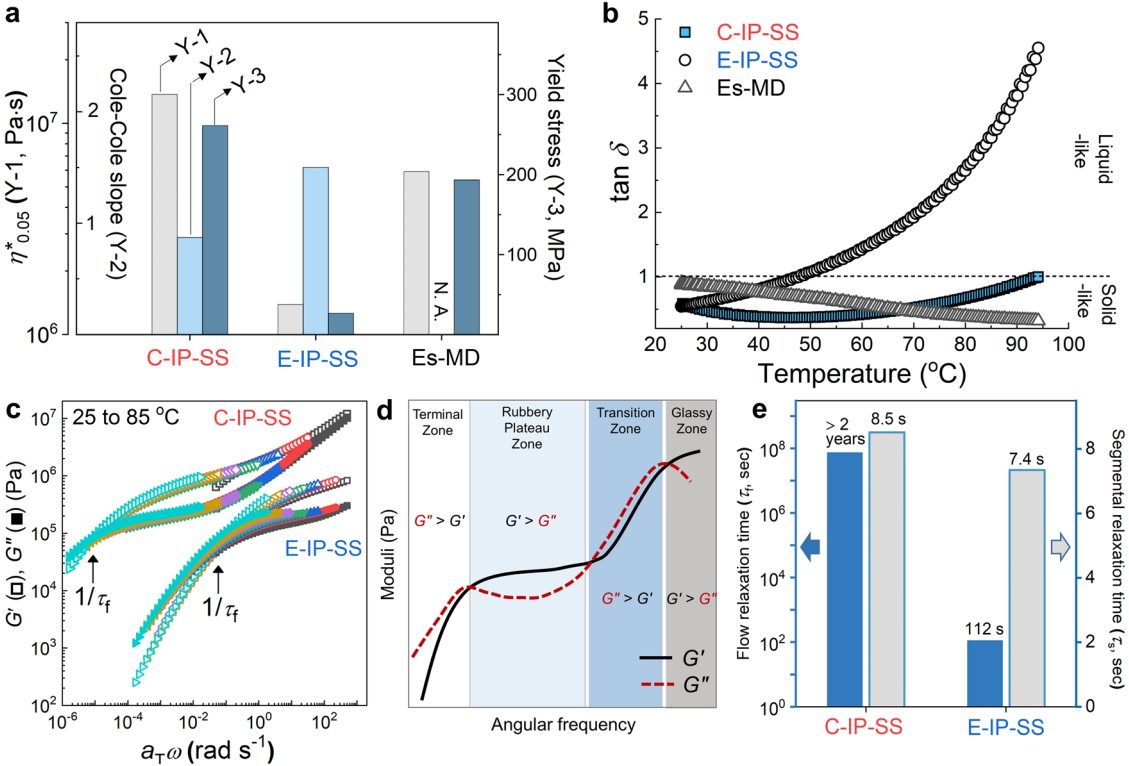

**Fig. 5 Rheological evidence for the self-healing mechanism. a** Rheological parameters for the three TPUs in static mode: complex viscosities ($\eta^\star_{0.05}$) were obtained from viscosity curves at 0.05 rad s$^{-1}$ (Supplementary Fig. 13a). Yield stresses were calculated from Casson plots (Supplementary Fig. S13b) and Cole–Cole plot slopes were obtained at 25 °C (Supplementary Fig. S13c). **b** Variations in loss tangent (tan $\delta$) over the 25–95 °C temperature range at 1 rad s$^{-1}$. The dashed line indicates tan $\delta = 1$. **c** Master curves of storage ($G'$) and loss ($G''$) moduli for C-IP-SS and E-IP-SS. The reference temperature is 25 °C. **d** Illustration depicting the representative $G'$ and $G''$ master curves. **e** Relaxation times for chain flow transition ($\tau_f$) and segmental motion ($\tau_s$) for C-IP-SS and E-IP-SS at 25 °C. The $\tau_f$ values were determined from the reciprocal values of the crossover frequencies of the $G'$ and $G''$ master curves (Fig. 5c), and the $\tau_s$ values were obtained from relaxation curves at 0.05 rad s$^{-1}$ that were calculated using Eq. (3) (Supplementary Fig. 14).

The key to C-IP-SS self-healing can be revealed from the segmental relaxation time ($\tau_s$), which was calculated from its relaxation time curve at an angular frequency ($\omega$) of 0.05 rad s$^{-1}$ (Supplementary Fig. 14). The $\tau_s$ of C-IP-SS was calculated to be 8.5 s, very similar to that of E-IP-SS (7.4 s) (Fig. 5e). C-IP-SS self-healing is therefore a consequence of high segmental vibrations that enable the exchange of disordered H-bonds, and dynamic SS in the poorly microphase-separated system. We also note that classical chain-flow-mediated self-healing degrades mechanical properties, while segmental motion-driven self-healing minimizes the mechanical-property penalty. The difference between the molecular dynamics of flow transition and segmental motion is schematically illustrated in Supplementary Fig. 15.

X-ray tools can be used to trace the structural changes that occur as TPUs transition from static to dynamic mode during stretching (Fig. 6a, b). The long- and short-range structural regularities of a TPU can be examined by small- and wide-angle X-ray scattering (SAXS and WAXS) techniques, respectively. To investigate static mode, we examined undrawn samples. While Es-MD presents characteristic SAXS patterns arising from clear phase separation (Supplementary Figs. 16 and 17), both C-IP-SS and E-IP-SS do not exhibit distinct SAXS patterns because they are poorly phase separated. When examined by WAXS, only Es-MD displayed a dim peak, which is evidence for the presence of crystalline hard segments (Fig. 6a and Supplementary Fig. 18).

To investigate dynamic mode, we analyzed strained translucent specimens. C-IP-SS and E-IP-SS still failed to show any SAXS peaks, indicating that external stress does not lead to phase separation. C-IP-SS showed the most noticeable change in its 1D WAXS spectrum when deformed at high strain (Fig. 6b), which suggests that C-IP-SS undergoes a distinct structural conversion into a crystalline form. Two peaks appeared at q values of 14.0 and 16.1 nm$^{-1}$ in the 1D WAXS spectrum when stretched to 400%; these peaks correspond to the (020) and (110) planes of crystalline methylene groups, respectively[63], with two clear spots observed in the equatorial region of the 2D WAXS pattern (Fig. 6a). Stretched carbonate-type soft segments have relatively regular and aligned secondary structures and therefore form metastable crystals in which methylene segments are laterally packed with the aid of the ordered H-bonding array (Supplementary Fig. 19). Consequently, C-IP-SS stiffens and becomes as strong as commercial Es-MD. On the other hand, E-IP-SS shows a strain-independent amorphous halo 2D WAXS pattern, and 400% elongation of Es-MD provides a clearer spot in the equatorial region of its 2D WAXS pattern and a small change in the 1D pattern compared to that prior to stretching. The hard domain is oriented along the stretching axis with minor crystallite growth. The minor amounts of strain-induced crystallization reveal that Es-MD and E-IP-SS behave like classic elastomers that observe almost isothermal entropic elasticity (i.e., $\Delta H$ or $\Delta U \approx 0$).

The 2D pattern of the relaxed C-IP-SS was acquired (Supplementary Fig. 20) and reveals that its metastable crystals had fully returned to the original amorphous phase upon unloading due to the instantaneous nature of its ordered H-bonding array. Meanwhile, the strain recovery of C-IP-SS degraded when held for a prolonged time (72 h) at 400% strain (Fig. 6a, b), which indicates that the metastable crystal structure is

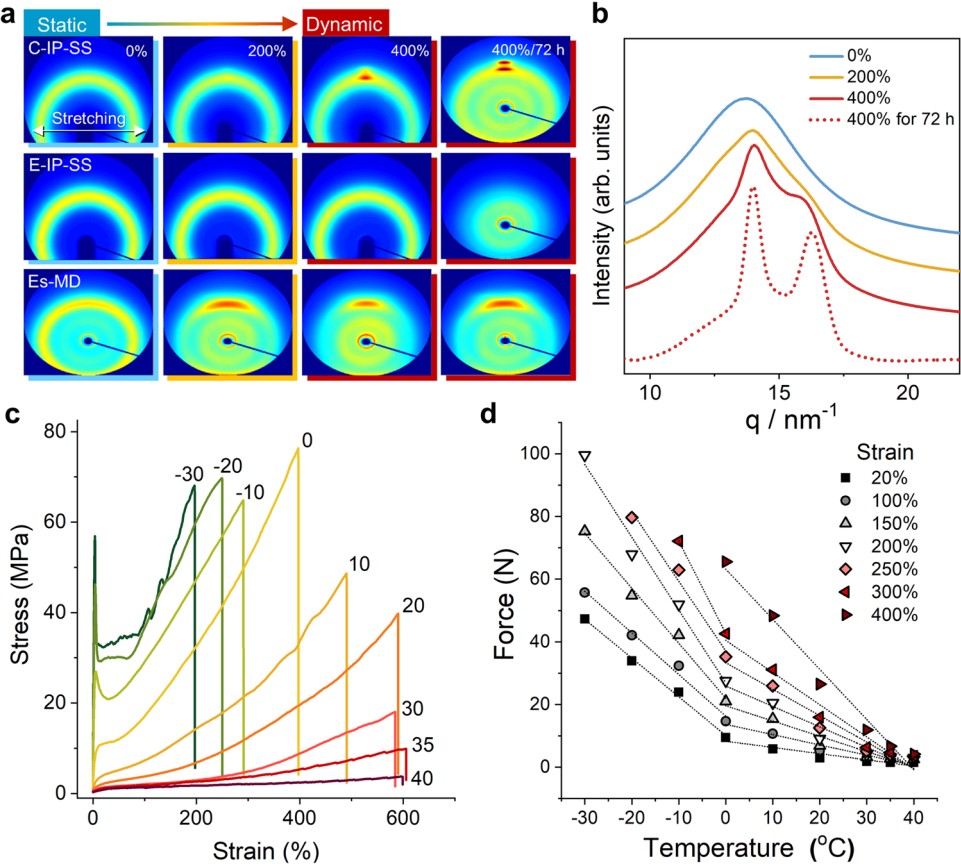

**Fig. 6 X-ray and thermodynamic analyses of strain-induced phase transition. a**, 2D WAXS images of three TPUs at stretching ratios of 0%, 200%, and 400%. The WAXS results of 400%-stretched samples and that were held for 72 h are also included to trace further structural evolution. **b**, 1D WAXS profiles of C-IP-SS at various degrees of stretching. **c** Stress–strain curves of C-IP-SS at various temperatures in the −30 to 40 °C range. **d** Force versus temperature plots at various strains based on the stress–strain curves.

stabilized upon long-term extension due to more organized lateral stacking and H-bonding.

**Practical calculation of reversible mechano-responsive phase conversion.** The full strain recovery of C-IP-SS is consistent with the phase-conversion-induced elasticity observed for biomaterials[64]. The egg-capsule wall of a channeled whelk was reported to exhibit highly reversible extensibility in which internal-energy return prevails over entropic relaxation upon unloading[65]. In other words, the reversible deformation of C-IP-SS is a non-isothermal process with an internal energy change that occurs through mechano-responsive crystallization. The internal energy contribution to the elasticity of C-IP-SS was evaluated on the basis of a well-established thermodynamic framework, as encapsulated by the following Eq. (1):

$$f = f_U + f_S = \left(\frac{\partial U}{\partial L}\right)_{T,V} - T\left(\frac{\partial S}{\partial L}\right)_{T,V} = \left(\frac{\partial U}{\partial L}\right)_{T,V} + T\left(\frac{\partial f}{\partial T}\right)_{L,V} \tag{1}$$

where $U$ and $S$ are the internal energy and entropy of the system, respectively, and $L$, $T$, and $V$ are the length, temperature, and volume of the specimen, respectively. The total elastic force ($f$) is the sum of the changes in $U$ and $S$ as functions of $L$ ($f_U$ and $f_S$) at constant $V$ and $T$[52]; $f_S$ is determined from an $f$ versus $T$ term through Maxwell's relationship. Figure 6c displays tensile stress–strain graphs of C-IP-SS at various temperatures between −30 and 40 °C, from which $f$ versus $T$ plots at particular strains (i.e., $L$) were constructed (Fig. 6d). A classic elastomer has a

positive slope because it is an isothermal system; hence $f_U = 0$ and $_S$ decreases with $L$, and $f_S < 0$, which leads to entropy-driven elasticity[29,66]. However, the plots for C-IP-SS have negative slopes at all examined $L$ and $T$ values, which means that stretching changes the internal energy (i.e., $f_U \neq 0$). The $U$ term prevails over the $S$ term for $f$. It is quite noticeable that the slope critically changes at 0 °C, which is close to the glass-transition temperature ($T_g$) (Supplementary Fig. 21). The steeper slope indicates that a higher internal-energy change results in more-stable and ordered crystals, which is associated with the following observations. C-IP-SS was drawn and released above its $T_g$; the original strain is then recovered because of its relatively unstable crystal phases. However, C-IP-SS is not reversible below $T_g$ because relatively stable crystals are formed. The energy ($\Delta U$) required to form the stress-induced metastable crystal is calculated to ~6.47 cal cm$^{-3}$ above $T_g$, the details of which are described in Supplementary Note 1 (Supplementary Figs. 22–24 and Supplementary Table 4). At a TPU density of 1.25 g cm$^{-3}$, this value can be converted into ~5.17 cal g$^{-1}$. The calculated heat of the metastable crystal is 3–10-times lower than those (15−65 cal g$^{-1}$) of stable commercial-polymer crystals[67], which qualitatively shows that the instantly formed strain-induced crystal is unstable.

**Biocompatibility Testing.** E-skin and wearable soft robots for healthcare mimic the nature of human body organs. Flexible and stretchable structural materials are essential for such purposes and have many requirements[13], including similar Young's moduli to those of soft tissue (10$^4$–10$^7$ Pa), mechanical robustness, self-healing, and biocompatibility. However, current elastomers

and hydrogels lack self-healing capabilities, which is a distinctive characteristic required to mimic natural organs. Therefore, this TPU is regarded as a suitable structural material for healthcare e-skin and soft robots due to the relevant combination of its exploitable stiffness, high mechanical performance, and damage-recovery.

As the last requirement, the physiological adaptability of C-IP-SS was evaluated by examining the in vitro cytotoxicities of C-IP-SS and commercial Es-MD in four types of cell, namely human breast carcinoma MDA-MB-231, human epidermoid carcinoma KB, Chinese hamster ovary normal CHO-K1, and mouse macrophage normal RAW 264.7, following the ISO 10993-5 guidelines (Supplementary Fig. 25)[68]. Cytotoxicity testing began with liquid extracts of the plastic materials. C-IP-SS was immersed in a cell growth medium with the plastic sample (area: 1 cm², thickness: 2 mm) in 1 mL of the medium at 36.5 °C for 72 h. The four types of cell were cultured in a complete growth medium, after which the culture medium was replaced with (1) neat cell culture medium (negative control), (2) cell culture medium with 20% (v/v) C-IP-SS extract (experimental group), or (3) cell culture medium with 5% dimethyl sulfoxide (DMSO) (positive control). The cells were then incubated for an additional 24 h and cell viabilities were evaluated using a colorimetric assay. Es-MD was tested using the same procedure. Both C-IP-SS and Es-MD exhibited consistently negligible cytotoxicity to the four types of mammalian cells; i.e., more than 80% of the cells were viable compared to the negative control[68,69].

The in vivo biocompatibility of C-IP-SS was evaluated by the Daegu Gyeongbuk Medical Innovation Foundation (DGMIF), a contract clinical research organization, following the ISO 10993-6 Annex A standard[69], and ethically approved by the Institutional Animal Care and Use Committee (Korea) (IACUC; approval code DGMIF-19042402-00) (Fig. 7a). As a positive control, the commercial TPU film (Es-MD) was immersed in DMSO for 12 h, and excess DMSO was gently wiped away. A high-density polyethylene (HDPE) film is a representative bio-inert material and was used as a negative control. All specimens were prepared as disk-shaped samples (diameter: 10 mm, thickness: 2 mm). For sterilization, they were sealed in packs and exposed to ethylene oxide gas for 12 h. The experimental animal model used male Sprague–Dawley rats (8-weeks-old, 250–300 g) (n = 4). Two experimental samples (C-IP-SS), one positive control and one negative control film, were implanted in four different subcutaneous regions (15 mm incisions) of a rat. The rats were euthanized after 12 weeks and the subcutaneous tissue samples were histopathologically analyzed after routine fixing and dyeing. Inflammatory responses were scored semi-quantitatively by a pathologist following the ISO 10993-6 inflammatory-reaction intensity guidelines: non-irritant < slight < moderate < severe. The existence, number, and distributions of polymorphonuclear cells, lymphocytes, plasma cells, macrophages, giant cells, and necrosis, and change in tissue caused by neovascularization, fatty infiltration, and fibrosis were evaluated. Supplementary Table 5 lists the scoring data, while Fig. 7b presents histopathological tissue images. C-IP-SS produced less inflammatory cells than the positive control. A small number of polymorphonuclear cells and lymphocytes were only observed in the tissue adjacent to C-IP-SS. The C-IP-SS film scored the lowest inflammatory intensity; i.e., non-irritant. In the preliminary experiments performed in this study, C-IP-SS showed favorable biocompatibility and did not elicit chronic or severe inflammation.

## Discussion

It is no exaggeration to state that self-healing polymers will transition toward commercialization if their mechanical performance meets industrial requirements. The uniquely designed carbonate-type macrodiol-based TPU developed in this study exhibited a tensile strength of 43 MPa, which is as high as industrially used elastomers with strong healing abilities. The key to the success of this TPU rests with its similarity to biological tissue; it operates in dual mode by responding to external stress through a reversible structural disorder-to-order transition of its H-bonding array. In static mode, the amorphous and disordered TPU matrix facilitates rapid segmental motion that promotes reversible self-healing bond exchange. Despite this self-healing behavior, this TPU, which contains a high-density internal H-bonding structure formed through its carbonate groups, delivered twice the elastic modulus of similar materials. This carbonate-type TPU exhibited strain-induced hardening and crystallization in dynamic mode due to a transformation toward the ordered mechano-responsive H-bonded array responsible for the strong and tough properties observed in the stretched state. In addition, as evidenced by X-ray, rheology, and temperature-dependent tensile studies, the major determining factor responsible for the full recovery of the phase transition is a non-isothermal process with an associated internal energy change of 5.17 cal g$^{-1}$, which is smaller than that associated with a stable polymeric crystal. Furthermore, its good biocompatibility highlights its promise for use in biomedical applications. We conclude that this mechano-responsive dual mode of operation captures both mechanical and self-healing performance at commercially relevant levels. In the next research, it might be necessary to perform fundamental study at the molecular level and also careful interpretation of H-bonding behavior in the self-healing TPU due to complex influences of other non-covalent bonds and polymer conformation.

## Methods

**Materials.** Poly(hexamethylene carbonate) diol (HPCD, $M_n = 1000$ g mol$^{-1}$) and commercial thermoplastic polyurethane (Es-MD, 5575AP grade) were kindly provided by the Dongsung Corp. (Korea). Isophorone diisocyanate (98%), N,N'-dimethylacetamide (DMAc, 99.8%), and dibutyltin dilaurate (DBTDL, 95%) were purchased from Sigma-Aldrich (USA), and bis(4-hydroxyphenyl)disulfide (98%) was purchased from TCI (Japan) and were used without further purification.

**TPU synthesis.** A typical polymerization procedure has been reported by us previously[47]. A typical polymerization procedure for C-IP-SS is described below. HPCD (C, 15.8 g, 15.8 mmol) was placed in a dried glass vessel equipped with a mechanical stirrer and heated in an oil bath at 100 °C under vacuum (<133 Pa) for 1 h to remove any moisture, after which it was cooled to 70 °C. Isophorone diisocyanate (IP, 7.38 g, 33.2 mmol) and DBTDL (0.050 g, 1800 ppm) dissolved in DMAc (8 mL) were added dropwise to the reaction vessel, with stirring continued

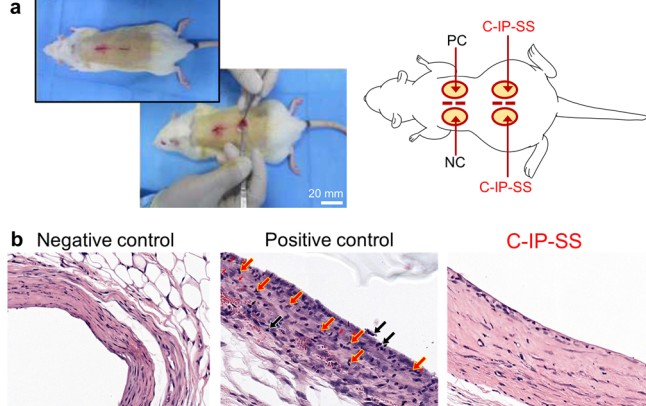

**Fig. 7 Biocompatibility test of C-IP-SS. a** Experimental procedure for in vivo histocompatibility testing: rat subcutaneous connective tissues with negative control (NC, HDPE), positive control (PC, DMSO-containing polyurethane), and the C-IP-SS film. **b** Representative histopathologic tissue images after 12 weeks of healing. Arrows indicate inflammatory cells (red: polymorphonuclear cells; black: lymphocytes).

for 2 h under a N$_2$ atmosphere. After the pre-polymer had been synthesized, the reactor was cooled to room temperature, and bis(4-hydroxyphenyl)disulfide (SS, 3.96 g, 15.8 mmol) dissolved in DMAc (15 mL) was added (as a chain extender) to the reactor. The reactor was heated to 40 °C and the reaction was continued until the NCO peak in the FT-IR spectrum of the mixture has disappeared, which required 1.5 h for C-IP-SS. For further characterization, the final concentration of the C-IP-SS solution was adjusted to 40 wt% by the addition of DMAc. M$_w$: 24,100 g mol$^{-1}$, PDI: 1.46. $^1$H NMR (CDCl$_3$, 300 MHz): δ = 7.37–7.27 (d, 4H), 6.84–6.71 (d, 4H), 4.17–4.07 (t, 24H), 4.07–3.97 (br, 4H), 2.96–2.80 (br, 4H), 1.79–1.51 (br, 40H), 1.49–1.30 (br, 28H), 1.12–0.71 (br, 18H).

**Film preparation, self-healing, and tensile testing.** A 40 wt% DMAc solution of TPU was poured onto a Teflon sheet and gradually heated from 60 to 100 °C over 48 h. Residual solvent was removed by vacuum drying at 110 °C for 12 h. A TPU film was prepared using a hot-press machine (Ocean Science Co., COAD1007b, Korea) and pressed into Teflon molds at 130 °C and 100 bar for 5 min. The pre-pared films were 100 mm × 100 mm × 1 mm in size. Mechanical and self-recovery properties were examined with a universal testing machine (UTM) (Instron 5943, USA), loaded with a 1 kN load cell and driven at a constant crosshead speed of 100 mm min$^{-1}$ at room temperature (25 °C). A 1-mm-thick dumbbell-shaped sample (ISO 37-4), with a gauge length of 12 mm and a width of 2 mm, was cut in half with a blade and then reattached to evaluate its self-healing properties. Temperature-dependent tensile testing at various temperatures between −30 and 40 °C was performed using a UTM (Instron 5982, USA) equipped with a temperature-controlled chamber. A 0.4-mm-thick dumbbell-shaped specimen (ASTM D638-5), with a gauge length of 9.53 mm and a width of 3.18 mm, was fabricated for temperature-dependence testing. The tensile test data of quintuplicate samples are expressed as mean ± the standard deviation.

**Structural characterization.** $^1$H NMR spectroscopy was conducted on an Advance 300 MHz spectrometer (Bruker, USA). The molecular weights and molecular-weight distributions of the polymers were determined by gel permeation chroma-tography (GPC) using ACQUITY APC XT columns (Waters, USA) at 40 °C on an instrument equipped with a refractive index detector, with tetrahydrofuran (THF) as the mobile phase. The M$_n$, M$_w$, and polydispersity index (PDI) were calculated relative to linear polystyrene standards. Attenuated total reflection (ATR)–FT-IR spectra were acquired on a Nicolet iS50 instrument (Thermo, USA). All TPU films were scanned 32 times at a resolution of 4 cm$^{-1}$. To ensure the data reliability and subsequently draw the 2D gradient map, the essential pre-treatment procedures of FT-IR data, including the baseline correction and normalization, were performed. The transmittance of each film was measured on a UV-2600 UV/vis spectrometer (SHIMADZU, Japan). DSC was performed on a Q-2000 instrument (TA Instruments, USA) between −90 and 180 °C in a N$_2$ atmosphere at a heating/cooling rate of 10 °C min$^{-1}$. One-dimensional wide- and small-angle X-ray scattering profiles were obtained using synchrotron radiation (λ = 1.28 Å) on the 3 C SAXS I beamline of the Pohang Accelerator Laboratory (PAL). The wavelength and pixel size were 1.229 and 0.079 mm, respectively. The sample to detector distance was 1.929 m for SAXS and 0.012 m for WAXS. Portable stretcher equipment was used to elongate the ~1-mm-thick dumbbell-shaped samples. Dynamic mechanical analysis (DMA) was performed using a Q800 instrument (TA Instruments, USA) at a fixed fre-quency of 1 Hz. The test specimens were bar-shaped films that were approximately 12.8 mm × 5.5 mm × 0.35 mm in size. Experiments were carried out at a heating and cooling rate of 3 °C min$^{-1}$ from −80 to 80 °C in a N$_2$ atmosphere.

**Rheology.** Rheological properties were examined on an oscillatory rheometer (MCR302; Anton Paar, Austria) using a 25 mm parallel plate-plate geometry. Prior to each experiment, approximately 0.5–0.9 mm thick film samples were prepared, and each film sample was placed between the parallel plates. A temperature-sweep experiment was carried out between 25 and 95 °C at an ω and shear strain of 1 rad s$^{-1}$ and 1%, respectively. In addition, frequency sweeping was carried out in the 0.05−500 rad s$^{-1}$ ω range at a constant shear strain of 1%. The yield stress, which provides the minimum energy required to collapse the internal structure, was obtained from a modified Casson plot using the Eq. (2) as follows[20]:

$$G''^{1/2} = G_y''^{1/2} + K\omega^{1/2} \qquad (2)$$

in which $G''_y$ is the yield stress and $K$ is a constant. The segmental relaxation behavior of each TPU was examined on the basis of the relaxation time obtained from the frequency-sweep data:

$$J' = G'/([\eta^*]\omega)^2 = \lambda/[\eta^*] \qquad (3)$$

in which, $J'$ is compliance, $G'$, $\eta^*$, and $\lambda$ are the storage modulus, complex viscosity, and relaxation time, respectively. The $\tau_s$ was obtained from the $\lambda$ value at 0.05 rad s$^{-1}$. To establish a time-temperature superposition (TTS) master curve, data were acquired at intervals of 10 °C in the 25–85 °C range. TTS master curves of $G'$ and $G''$ for E-IP-SS and C-IP-SS were constructed from the frequency-sweep data by shifting the data to the reference temperature (25 °C).

The frequency shift factor, α$_T$, was determined as follows:

$$\alpha_T = \frac{\omega(T)}{\omega(T_0)}$$

$$G'(T,\alpha_T\omega) = G'(T_0,\omega)$$

$$G''(T,\alpha_T\omega) = G''(T_0,\omega) \qquad (4)$$

The value of α$_T$ obtained at each temperature is summarized in Supplementary Table 6.

**Biocompatibility testing.** In vitro cytotoxicity testing was performed according to the ISO 10993-5 international standard[69]. Cytotoxicity experiments began with liquid extracts of the plastic materials. A 1 cm$^2$ C-IP-SS film sample sterilized with 70% aqueous ethanol solution was immersed in 1 mL of cell growth medium at 36.5 °C for 72 h. The culture medium extract was filtered through a syringe filter. Human breast carcinoma MDA-MB-231 cells, human epidermoid carcinoma KB cells, normal Chinese hamster ovary CHO-K1 cells, and normal mouse macro-phage RAW 264.7 cells were purchased from the Korea Cell Line Bank and cul-tured in a Dulbecco's modified Eagle's medium (DMEM) supplemented with 10% fetal bovine serum, 100 U mL$^{-1}$ penicillin G, 100 μg mL$^{-1}$ streptomycin, and 0.025 μg mL$^{-1}$ amphotericin B at 36.5 °C for 24 h in 5% CO$_2$ atmosphere. During in vitro cytotoxicity testing, the culture medium was replaced with neat culture medium (negative control), culture medium with 20% (v/v) C-IP-SS extract, or culture medium with 5% DMSO (positive control). The cells were then incubated for an additional 24 h to expose the cell to the extract. The CCK-8 cell counting kit (Dojindo, Inc., Rockville, MD, USA) was used to evaluate cell viability. Incubated media were transferred to fresh 96-well plates for colorimetric assessment at 450 nm using a microplate reader. Es-MD cytotoxicity testing followed the same procedure. An absorbance intensity that corresponds to less than 80% cell viability compared to the negative control indicates that the sample is not biocompatible. Data from quintuplicate samples are expressed as means ± standard deviations. All data were evaluated using a Student's t-test at a not significance (NS) of $p > 0.05$ and a significance of $p < 0.01$ (**).

All surgical procedures for in vivo biocompatibility testing were conducted by Daegu Gyeongbuk medical innovation foundation (DGMIF) (http://www.dgmif.re.kr/eng/index.do), a public contract clinical research organization, with the approval of the national institutional review board (IRB). The ethical issue for animal testing was approved by IACUC (Korea) with the approval code of DGMIF-19042402-00. The samples were implanted in male Sprague–Dawley rats (8-weeks-old, 250–300 g) (n = 4). The rats were allowed free access to food and water in the room under the temperature- and humidity-controlled conditions (22 °C, 50%) with a 12/12 h day/night cycle (8 am/8 pm). Each rat was anesthetized using an intramuscular injection of 50 mg ml$^{-1}$ Zoletil 50 (tiletamine and zolazepam; Virbac, Carros, France) and 23 mg ml$^{-1}$ Rompun (xylazine; Bayer, Leverkusen, Germany) and, the incision of the scalp proceeded carefully. Two experimental samples (C-IP-SS), one positive control (DMSO-TPU), and one negative control (HDPE) films (10 mm diameter circle) were implanted in four different regions (15 mm incision) of subcutaneous rat tissues. The incised skins were closed with 4/0 Dafil sutures (Ethicon, Somerville, NJ, USA) and disinfected with povidone after whole procedures. After the surgery, the rats were maintained in their cages for 12 weeks, and then, the rats were sacrificed for histological analyses.

The tissue samples were routinely dehydrated, paraffin-embedded, cut, and stained with hematoxylin and eosin (H&E). Then, the tissue cross-sections were examined for the semi-quantitative evaluation regarding the International Standard (ISO 10993-6, Annex A) criteria for the biological evaluation of the local effects of medical devices after implantation by a pathologist. The local effects were estimated by comparing the tissue response caused by the experimental samples (C-IP-SS) and the negative control. The scoring system follows the histological evaluation of the extent of the area affected. The existence, number, and distribution of polymorphonuclear cells, lymphocytes, plasma cells, macrophages, giant cells, and necrosis were evaluated. Changes in the tissue caused by neovascularization, fatty infiltration, and fibrosis were evaluated as well.

**Reporting summary.** Further information on research design is available in the Nature Research Reporting Summary linked to this article.

## Data availability

The source data that support the findings of this study are available (https://doi.org/10.6084/m9.figshare.12936989). We provide the source data underlying Figs. 1b, 1c, 3a, 4a, 4b, 5a, 5b, 5c, 5e, and 6b–d, and Supplementary Figs. 2–6, 9–11, 13, 14, 16, 18, 21, and 23–25. Source data are provided with this paper.

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

## Acknowledgements

This research was supported by KRICT (Core Project; SS2042-10, and Excellence Research Group Project; BSF20-254) and the National Research Foundation of Korea (NRF) funded by the Ministry of Science, ICT & Future Planning (2018R1C1B6000966, 2019R1C1C1003888, and 2020R1C1C1009340). We acknowledge the Pohang Accelerator Laboratory for X-ray diffraction experiments with synchrotron radiation (Beamline 3 C).

## Author contributions

Y.E., S.-M.K., M.L., and H.J. performed the experiments and analyzed the data. Y.E., S.-M.K., J.P., and D.X.O. prepared the figures and wrote the paper. Y.E. created and drew schematic illustrations. S.Y.H., J.P., and D.X.O. conceived, designed, and directed the project. J.P. (CUK) and E.S.L. performed the in vitro cell experiments. All authors have given approval to the final version of the manuscript.

## Competing interests

The authors declare no competing interests.
