## [Peer Review File · Nature Communications]

REVIEWER COMMENTS

Reviewer #1 (Remarks to the Author):

The manuscript entitled “Mechano-responsive hydrogen-bonding array of thermoplastic polyurethane elastomer captures both strength and self-healing” comprise details the development of a self-healable carbonate-type thermoplastic polyurethane at 35 °C; which they claim to be stronger than conventional footwear elastomers. The carbonated-polymer (C-IP-SS) presented in this manuscript builds on a polymer base (IP-SS) developed by the same group and previously published (ref 47). Also, they are bringing the ether-polymer (E-IP-SS) as a control polymer in the current study. The manuscript presents a relevant composition and some innovativeness despite previous publication from the group resembling the system described herein. Despite a comprehensive materials characterization, some of the biological test could also have been more detailed and better presented for the reader. I therefore recommend that the authors could resubmit the manuscript to this journal or another journal after addressing the below mentioned queries:

1) On page 8, from line 139 to 141, it was mentioned “Inspired by the recovery process of biomaterials, we deduce that rapid switching between the two individual internal structures drives the reversible deformation of C-IP-SS (Fig. 2d)”, and they added reference 48. Here, I suggest the author to expand their discussion by explaining how reference 48 exactly has driven them to this conclusion, instead of only referring to it as they currently are doing.

2) Figure 5d shows that the stress vs. strain curves for C-IP-SS polymer at 20 °C, 30 °C and 35 °C do not correspond or are not similar to the results presented in Figure 1b and Figure 1c for the same material. I recommend the authors to explain the inconsistency in these results.

3) The biological results in Figure 6 are merely qualitative. The biological results could be better presented, proved and discussed via statistical analysis by comparing with control materials.

4) The innovativeness of the application of this paper would be more obvious if Suppl. Figure 7 as moved into the main text instead.

Yours sincerely,
Alireza Dolatshahi-Pirouz

Reviewer #2 (Remarks to the Author):

In this manuscript, the authors incorporate hydrogen bond acceptors to the backbone of a polymer chain to achieve strain induced crystallization in a self-healing polymeric system. In its unstrained and low strained states, the self-healing elastomer behaves like a common physically crosslinked

polymer network, the elasticity of which mostly comes from the entropy of polymer strands. In high strained states, the crystallites are formed in the elastomer due to the high strain, and this crystallization improves the ultimate tensile strength and toughness. Strain induced crystallization is not a new strategy for enhancing the mechanical properties of an elastomer, but introducing it into a self-healing matrix is an interesting idea. I take it at face value that the mechanical properties of this self-healing elastomer are outliers – I find that such data are difficult to search for, but I am not aware of an exception – and for that reason I might reconsider recommending this work to be published on Nature Comm, after the authors address my concerns. In particular, I worry that much of the analysis remains somewhat speculative and that some aspects of the mechanism are unknown. More importantly, but related to this, it is not clear how one would take the analysis and use it to inform future designs, other than in the most general sense. So the impact of the paper truly lies with just how remarkable the reported properties are.

Below are several points that were found to be somewhat confusing, and I hope they will help to improve the manuscript.

1. Throughout the paper, the authors mention “soft domain” and “hard domain”. It makes some sense when the authors are discussing the traditional TPU, which has the crystallized MD moieties as the crosslinkers, which can be intuitively considered as “hard”. However, the manuscript also said the SS and IP moieties are not crystallized at unstrained state. So why are they “hard”? Can the author define what they mean by “soft” and “hard”?
2. Can the authors specify what holds the network together in C-IP-SS elastomer, and furnishes the memory of the initial state? Is it just the hydrogen bonding between carbamate groups and carbamate/carbonate groups? Why do the effects persist longer than the strain induced crystallization?
3. In line 92, the author mentioned the Mn of both TPUs are 1kDa, but the SI shows the Mn to be 16.5kDa and 30.4kDa.
4. The authors should explain more about how they assign the peaks of IR for the C-IP-SS.
5. From line 162 to 165, there are numerous grammatical mistakes.
6. In Figure 3b, why does the peak at 1692cm^{-1} increase at low strain?
7. In Figure 3b, why does the peak at 1650cm^{-1} increase at high strain?
8. In Figure 3b, it looks like that the peaks at 1718 , 1692 , 1645 cm^{-1} all increase at high strain, and the peak at 1741 cm^{-1} decreases. It looks like the H-bonded C=O groups increase in number as strain increases. However, the amount of hydrogen bond donors limits the number of hydrogen bonds in the elastomer. Are those hydrogen bond donors just partially bonded at unstrained state? Can the authors show how the exact IR absorption changes instead of the derivative? If we look at Figure 5c, the modulus decreases significantly from 0 to 40 degree. It seems that the hydrogen bonding is the main component that holds the network together. So, at 10 degree, I would expect that most of hydrogen donors are bonded. Then why do the authors observe an increase in bonded C=O groups when stretching the elastomer?
9. Also, it is important to be sure that the change in sample geometry during straining does not influence any of the changes in intensity that are recorded.
10. In Figure 3c, I do not think there is a strong interaction between carbonate groups, at least not that is comparable in strength to the hydrogen bonding. The authors put this interaction together with other hydrogen bonding, which is confusing. This requires either a reference or stronger independent justification.
11. When calculating the energy required to form the crystallite, have the authors considered that

the crosslinking density is also changing? And the material is not identical at temperatures above T_g .

Reviewer #3 (Remarks to the Author):

This manuscript reports a novel thermoplastic polyurethane elastomer that displays an improved combination of strength and self-healing capacity. The strength of the manuscript relates to the mechanistic part where the fundamental aspects of the reversible strain-induced crystallization are unraveled. This part of the publication is designed, performed and interpreted at high scientific level. Nevertheless, the novelty and quality of the biological evaluation is rather limited.

As compared to previous work on self-healing polyurethanes (e.g. reference 47 by the same group) the improvement of the tensile strength should be regarded as incremental optimization rather than a breakthrough development that will create entirely new avenues of research. Moreover, the enhanced strength goes at the expense of ductility. Unfortunately, the authors only discuss the positive outcome of this study (improved strength and toughness) without properly discussing the negative results (reduced ductility as shown in Figure 1c).

Regarding the biocompatibility studies it should be stressed that the authors do not properly describe the intended application (i.e. muscle tissue replacement) in the introduction. They introduce the intended application only at the end of the manuscript (p17) by defining the concept of "precursor material for artificial muscle", but this concept is not defined adequately from a biological/clinical perspective. What is a "precursor material for artificial muscle", and why would the newly developed TPU be a suitable candidate biomaterial? Why are the improved strength and toughness vs. reduced ductility of the newly developed TPU relevant for muscle tissue replacement?

The *in vivo* implantation study itself is qualitative at best and does not provide sufficient evidence to justify the authors' claims on superior biocompatibility and reduced inflammatory responses. The authors should provide more quantitative evidence (e.g. counting of inflammatory cells or staining of other inflammatory markers) to justify these claims. The selected implantation model does not allow to draw conclusions on the functionality of the implanted TPU, while the authors should have selected an intramuscular instead of subcutaneous implantation site to confirm the suitability of the developed biomaterial for muscular replacement. The authors should consider to compare the biological performance of their novel C-IP-SS polyurethane with conventional non-selfhealing polyurethanes (such as E-IP-SS and Es-MD shown in Figure 1) to confirm the superiority of self-healing polyurethanes over conventional polyurethanes. To provide further evidence for the cytocompatibility of the newly developed materials, *in vitro* cell culture studies should be performed as well.

Finally, statistical analysis is required for the presented mechanical and biological tests in Figures 1, 2, 4 and 6, but this analysis is lacking unfortunately. Consequently, reproducibility of the results is currently questionable.

Minor details:

- Figure caption 6: "in vivo cytotoxicity" should be "in vivo histocompatibility" since the figure shows

tissue response rather than cellular response.

- Details on sterilization of implanted materials are lacking.

Point-to-point responses to referees

Contents

Responses to reviewer #1: 2 Page

Responses to reviewer #2: 12 Page

Responses to reviewer #3: 24 Page

Reviewers' comments in italic.

Responses highlighted in gray.

The updated contents highlighted in yellow

The dragged contents from the original manuscript in sky-blue color.

The materials only for revision in purple color.

[Response to Reviewer #1]

Overall comment: *The manuscript entitled “Mechano-responsive hydrogen-bonding array of thermoplastic polyurethane elastomer captures both strength and self-healing” comprise details the development of a self-healable carbonate-type thermoplastic polyurethane at 35 °C; which they claim to be stronger than conventional footwear elastomers. The carbonated-polymer (C-IP-SS) presented in this manuscript builds on a polymer base (IP-SS) developed by the same group and previously published (ref 47). Also, they are bringing the ether-polymer (E-IP-SS) as a control polymer in the current study. The manuscript presents a relevant composition and some innovativeness despite previous publication from the group resembling the system described herein. Despite a comprehensive materials characterization, some of the biological test could also have been more detailed and better presented for the reader. I therefore recommend that the authors could resubmit the manuscript to this journal or another journal after addressing the below mentioned queries:*

Response: All authors acknowledge and appreciate the reviewer's positive evaluation of our manuscript. We welcome the opportunity to address the issue raised in the reviewer’s report and make our manuscript more persuasive.

Comment 1: *On page 8, from line 139 to 141, it was mentioned “Inspired by the recovery process of biomaterials, we deduce that rapid switching between the two individual internal structures drives the reversible deformation of C-IP-SS (Fig. 2d)”, and they added reference 48. Here, I suggest the author to expand their discussion by explaining how reference 48 exactly has driven them to this conclusion, instead of only referring to it as they currently are doing.*

Response to Comment 1: We appreciate the reviewer’s detailed and critical comments. According to this suggestion, we have added a description of the phenomenological similarity of the elastic deformation between biomaterials and the synthetic self-healing TPU (C-IP-SS).

In conventional elastomers and rubbers, it is well-recognized that the elastic recovery upon stress is driven by the conformational entropy spring of chains with negligible phase transition. However, the reversible strain-induced crystallization of C-IP-SS reveals that the phase transition is intimately involved in the reversible deformation and related elastic

recovery. To understand the unique reversible deformation of C-IP-SS accompanied by the strain-induced phase transition, we focused on the elastic deformation behavior of natural elastomeric materials.

Similar behavior has been reported for biomaterials [*Chem. Soc. Rev.* **42**, 1973-1995 (2013)]. Some protein-based bioelastomers undergo reversible phase transition of the α -helix upon release of the β -sheet with deformation. The internal-energy change caused by the phase transition drives the reversible deformation. It is called "phase transition-induced elasticity." Therefore, we conjecture that the instant switching of the amorphous-to-crystalline phases of C-IP-SS is a major driving force for reversible deformation. For clarity, we have described in more detail how the deformation behaviors of the bioelastomers and the synthesized TPU are related.

- Line 145-149, "The strain-induced phase transition (α -helix \leftrightarrow β -sheet) in some natural elastomeric materials is involved in their reversible deformation upon stretching, which results in long-range elastic recovery.⁵² In other words, the internal energy change through this phase transition is a primary driving force for elasticity as opposed to the entropic behavior of a conventional rubber or elastomer."

Comment 2: *Figure 5d shows that the stress vs. strain curves for C-IP-SS polymer at 20 °C, 30 °C and 35 °C do not correspond or are not similar to the results presented in Figure 1b and Figure 1c for the same material. I recommend the authors to explain the inconsistency in these results.*

Response to Comment 2: We agree with the lack of a detailed explanation for the discrepancy of the stress–strain results in the two figures. As described in the updated Methods section, different types of universal testing machines (UTMs) were used in two different tests for 1) self-healing efficiency (Figure 1b) and 2) temperature dependence (previously Figure 5d, Figure 6c in the revised manuscript). The different UTMs require different specimen dimensions, which significantly affect the tensile results.

In the self-healing test, a single-column Instron 5943 model UTM and its grip jaw were deemed suitable for precisely evaluating low-load specimens. For example, 1 h of self-healed C-IP-SS and E-IP-SS only achieved low tensile strengths of 5–10 MPa. Unfortunately, the

single-column UTM cannot support the temperature-controlled chamber owing to the limited machine geometry. Hence, a double-column Instron 5982 UTM equipped with a temperature-controlled chamber was optimized for the temperature-dependent tensile properties.

Owing to the difference in UTMs, tensile-tested specimens with different dimensions were prepared in the two tests. As shown in Figure R1, the two different tests for self-healing efficiency and temperature dependency were performed based on ISO 37-4 and ASTM D638-5, respectively, and required different specimen sizes. Nonetheless, there is a minor discrepancy in property values obtained from the two different experimental conditions, e.g., the tensile strength (43 MPa) of C-IP-SS of Figure 1 \approx that (40–50 MPa) of C-IP-SS at 10–20 °C Figure 6.

To prevent misleading, more detailed information on the specimen dimensions in the two different tensile tests has been added as follows.

- Line 445-451, "A 1-mm-thick dumbbell-shaped sample (ISO 37-4), with a gauge length of 12 mm and a width of 2 mm, was cut in half with a blade and then reattached to evaluate its self-healing properties. Temperature-dependent tensile testing at various temperatures between -30 and 40 °C was performed using a UTM (Instron 5982, USA) equipped with a temperature-controlled chamber. A 0.4-mm-thick dumbbell-shaped specimen (ASTM D638-5), with a gauge length of 9.53 mm and a width of 3.18 mm, was fabricated for temperature-dependence testing."

Figure R1. Tensile testing machines and respective specimen information in (a) self-healing (ISO 37-4) and (b) temperature-dependency measurements (ASTM D638-5). The machine

images were provided from <https://www.instron.co.kr/products/testing-systems/universal-testing-systems>.

Comment 3: The biological results in Figure 6 are merely qualitative. The biological results could be better presented, proved and discussed via statistical analysis by comparing with control materials.

Response to Comment 3: As you suggested, we have updated the precise details of the results of the *in vivo* biological tests in the Supplementary Information. Semi-quantitative histocompatibility evaluation was conducted by a pathologist in a contract clinical research organization regarding ISO 10993-6. In addition, to exhibit more quantitative cytotoxicity of the samples, we conducted *in vitro* tests as follows.

In vivo test

- (Line 341-362) The *in vivo* biocompatibility of **C-IP-SS** was evaluated by the Daegu Gyeongbuk Medical Innovation Foundation (DGMIF), a contract clinical research organization, following the ISO 10993-6 Annex A standard⁶⁹, and ethically approved by the Institutional Animal Care and Use Committee (Korea) (IACUC; approval code DGMIF-19042402-00) (Fig. 7a). As a positive control, the commercial TPU film (**Es-MD**) was immersed in DMSO for 12 h, and excess DMSO was gently wiped away. A high-density polyethylene (HDPE) film is a representative bio-inert material and was used as a negative control. All specimens were prepared as disk-shaped samples (diameter: 10 mm, thickness: 2 mm). For sterilization, they were sealed in packs and exposed to ethylene oxide gas for 12 h. The experimental animal model used male Sprague–Dawley rats (8-weeks-old, 250–300 g) (n = 4). Two experimental samples (**C-IP-SS**), one positive control and one negative control film, were implanted in four different subcutaneous regions (15 mm incisions) of a rat. The rats were euthanized after 12 weeks and the subcutaneous tissue samples were histopathologically analyzed after routine fixing and dyeing. Inflammatory responses were scored semi-quantitatively by a pathologist following the ISO 10993-6 inflammatory-reaction

intensity guidelines: non-irritant < slight < moderate < severe. The existence, number, and distributions of polymorphonuclear cells, lymphocytes, plasma cells, macrophages, giant cells, and necrosis, and change in tissue caused by neovascularization, fatty infiltration, and fibrosis were evaluated. Supplementary Table 5 lists the scoring data, while Fig. 7b presents histopathological tissue images. C-IP-SS produced less inflammatory cells than the positive control. A small number of polymorphonuclear cells and lymphocytes were only observed in the tissue adjacent to C-IP-SS. The C-IP-SS film scored the lowest inflammatory intensity; i.e., non-irritant. In the preliminary experiments performed in this study, C-IP-SS showed favorable biocompatibility and did not elicit chronic or severe inflammation.

(In Supplementary Information) **Supplementary Table 5.** Average score of local irritation reaction to subcutaneous implants from histological data (n = 4)

Cell type/ tissue response	Experimental (C-IP-SS)	Control	
		Negative (HDPE)	Positive (DMSO-TPU)
Inflammation			
polymorphonuclear cells ^a	0.75	0.29	3.00
Lymphocytes ^a	0.25	0.14	2.50
Plasma cells ^a	0.00	0.00	0.00
Macrophages ^a	0.00	0.00	0.25
Giant cells ^a	0.00	0.00	0.00
Necrosis ^b	0.00	0.00	2.25
A: subtotal			
inflammation score (×2)	2.00	0.86	16.00
Neovascularization^c			
Fibrosis ^d	1.00	1.29	2.75

Fatty infiltrate ^e	0.00	0.00	0.25
B: tissue response subtotal	1.00	1.86	5.25
Total (A+B)	3.00	2.72	21.25
Score (= experimental – negative control) ^f	0.29	-	18.53
Traumatic necrosis	0	0	0
Foreign debris	0	0	0
No. sites examined	8	8	8

^a Mean histopathological score of reaction based on the number of cells per high-powered field at 400x: 0 = 0 cells; 1 = 1-5 cells; 2 = 5-10 cells; 3 = heavy infiltrate; 4 = packed.

^b 0 = None; 1 = minimal; 2 = mild; 3 = moderate; 4 = severe.

^c 0 = None; 1 = minimal capillary proliferation, focal, 1–3 buds; 2 = groups of 4–7 capillaries with supporting fibroblastic structures; 3 = broad band of capillaries with supporting structures; 4 = extensive band of capillaries with supporting fibroblastic structures.

^d 0 = None; 1 = narrow band; 2 = moderately thick band; 3 = thick band; 4 = extensive band.

^e 0 = None; 1 = minimal amount of fat associated with fibrosis; 2 = several layers of fat and fibrosis; 3 = elongated and broad accumulation of fat cells about the implant site; 4 = extensive fat completely surrounding the implant.

^f Non-irritant (0.0 up to 2.9) < slight (3.0 up to 8.9) < moderate (9.0 up to 15.0) < severe (more than 15.0).

The local irritation effects were evaluated by a comparison of the tissue response caused by the tested implant to that caused by the negative control. The stained tissue cross-sections were examined by a pathologist in a public contract clinical research organization using an optical microscopy at high-powered field of 400x, and the pathologist scored local irritation responses based on the number of cells and the tissue change. The all scores are average values (n = 4). The subtotal inflammation score was multiplied by 2 and added to the subtotal score of neovascularization, fibrosis, and fatty infiltrate. Then, the value was subtracted by the score of negative control. According to the final score, the sample can be considered as

non-irritant (0.0 up to 2.9), slight (3.0 up to 8.9), moderate (9.0 up to 15.0), and severe (more than 15.0).

In vitro test

Supplementary Figure 24. *In vitro* cell viability tests of negative control (pristine cell culture medium), positive control (cell culture medium with 5% DMSO), C-IP-SS extracts, and Es-MD extracts using four different types of cells: (a) Chinese hamster ovary normal CHO-K1, (b) human breast carcinoma MDA-MB-231, (c) human epidermoid carcinoma KB, and (d) mouse macrophage normal RAW 264.7. The data of quintuplicate samples are expressed as mean \pm the standard deviation. All data were evaluated using a Student's t-test at

a not significance (NS) of $p > 0.05$ and a significance of $p < 0.01$ (**).

- (Line 327-340): As the last requirement, the physiological adaptability of **C-IP-SS** was evaluated by examining the *in vitro* cytotoxicities of **C-IP-SS** and commercial **Es-MD** in four types of cell, namely human breast carcinoma MDA-MB-231, human epidermoid carcinoma KB, Chinese hamster ovary normal CHO-K1, and mouse macrophage normal RAW 264.7, following the ISO 10993-5 guidelines (Supplementary Fig. 24)⁶⁸. Cytotoxicity testing began with liquid extracts of the plastic materials. **C-IP-SS** was immersed in a cell growth medium with the plastic sample (area: 1 cm², thickness: 2 mm) in 1 mL of the medium at 36.5 °C for 72 h. The four types of cell were cultured in a complete growth medium, after which the culture medium was replaced with (1) neat cell culture medium (negative control), (2) cell culture medium with 20% (v/v) **C-IP-SS** extract (experimental group), or (3) cell culture medium with 5% dimethyl sulfoxide (DMSO) (positive control). The cells were then incubated for an additional 24 h and cell viabilities were evaluated using a colorimetric assay. **Es-MD** was tested using the same procedure. Both **C-IP-SS** and **Es-MD** exhibited consistently negligible cytotoxicity to the four types of mammalian cells; i.e., more than 80% of the cells were viable compared to the negative control^{68,69}.

- (Line 496-514): *In vitro* cytotoxicity testing was performed according to the ISO 10993-5 international standard⁶⁹. Cytotoxicity experiments began with liquid extracts of the plastic materials. A 1 cm² **C-IP-SS** film sample sterilized with 70% aqueous ethanol solution was immersed in 1 mL of cell growth medium at 36.5 °C for 72 h. The culture medium extract was filtered through a syringe filter. Human breast carcinoma MDA-MB-231 cells, human epidermoid carcinoma KB cells, normal Chinese hamster ovary CHO-K1 cells, and normal mouse macrophage RAW 264.7 cells were purchased from the Korea Cell Line Bank and cultured in a Dulbecco's modified Eagle's medium (DMEM) supplemented with 10% fetal bovine serum, 100 U mL⁻¹ penicillin G, 100 µg mL⁻¹ streptomycin, and 0.025 µg mL⁻¹ amphotericin B at 36.5 °C for 24 h in 5% CO₂ atmosphere. During *in vitro* cytotoxicity testing, the

culture medium was replaced with neat culture medium (negative control), culture medium with 20% (v/v) C-IP-SS extract, or culture medium with 5% DMSO (positive control). The cells were then incubated for an additional 24 h to expose the cell to the extract. The CCK-8 cell counting kit (Dojindo, Inc., Rockville, MD, USA) was used to evaluate cell viability. Incubated media were transferred to fresh 96-well plates for colorimetric assessment at 450 nm using a microplate reader. Es-MD cytotoxicity testing followed the same procedure. An absorbance intensity that corresponds to less than 80% cell viability compared to the negative control indicates that the sample is not biocompatible. Data from quintuplicate samples are expressed as means \pm standard deviations. All data were evaluated using a Student's t-test at a not significance (NS) of $p > 0.05$ and a significance of $p < 0.01$ (**).

- 68. Loh, X. J., Sng, J. B. C. & Li, J. Synthesis and water-swelling of thermo-responsive poly(ester urethane)s containing poly(ϵ -caprolactone), poly(ethylene glycol) and poly(propylene glycol). *Biomaterials* **29**, 3185-3194 (2008).

Comment 4: The innovativeness of the application of this paper would be more obvious if Suppl. Figure 7 as moved into the main text instead.

Response to Comment 4: According to the reviewer's suggestion, we have moved Supplementary Figure 7 (ashby plot) to the main text as Fig. 2 in the revised manuscript.

Fig. 2. Ashby plot of “ultimate tensile strength” versus “self-healing temperature” of C-IP-SS and other elastomers reported in literature.

[Response to Reviewer #2]

Overall comment: *In this manuscript, the authors incorporate hydrogen bond acceptors to the backbone of a polymer chain to achieve strain induced crystallization in a self-healing polymeric system. In its unstrained and low strained states, the self-healing elastomer behaves like a common physically crosslinked polymer network, the elasticity of which mostly comes from the entropy of polymer strands. In high strained states, the crystallites are formed in the elastomer due to the high strain, and this crystallization improves the ultimate tensile strength and toughness. Strain induced crystallization is not a new strategy for enhancing the mechanical properties of an elastomer, but introducing it into a self-healing matrix is an interesting idea. I take it at face value that the mechanical properties of this self-healing elastomer are outliers – I find that such data are difficult to search for, but I am not aware of an exception – and for that reason I might reconsider recommending this work to be published on Nature Comm, after the authors address my concerns. In particular, I worry that much of the analysis remains somewhat speculative and that some aspects of the mechanism are unknown. More importantly, but related to this, it is not clear how one would take the analysis and use it to inform future designs, other than in the most general sense. So the impact of the paper truly lies with just how remarkable the reported properties are. Below are several points that were found to be somewhat confusing, and I hope they will help to improve the manuscript.*

Response to Comment: We are delighted to have received positive comments from the reviewer. We have addressed all issues in the below comments and revised some main texts and figures to minimize speculation and confusion. We hope that the revised version is deemed acceptable by the journal.

Comment 1: *Throughout the paper, the authors mention “soft domain” and “hard domain”. It makes some sense when the authors are discussing the traditional TPU, which has the crystallized MD moieties as the crosslinkers, which can be intuitively considered as “hard”. However, the manuscript also said the SS and IP moieties are not crystallized at unstrained state. So why are they “hard”? Can the author define what they mean by “soft” and “hard”?*

Response to Comment 1: We strongly agree with the reviewer’s opinion that the non-

crystalline IP and SS cannot form the "hard domain." There has been a slight misunderstanding. We must distinguish between "domain" and "segment"

It is well-known that the chain structure of thermoplastic elastomers is composed of hard and soft segments. The terms "hard" and "soft" are determined with respect to the segment chain rigidity. The hard segment with high molecular rigidity mainly contains the aromatic or aliphatic ring moiety, whereas the soft segment with high molecular flexibility possesses methylene groups.

In typical thermoplastic elastomers, hard and soft segments develop the crystalline hard domain and amorphous soft domain through microphase separation.

In the current study, the carbonate chain of C-IP-SS is clearly a soft segment with respect to its low T_g . In this regard, the IP-SS moiety of our TPU is the hard segments in respect of the domain structure, i.e., aromatic and alicyclic structure. In contrast, C-IP-SS does not create both domains because of its poor microphase separation. The poor packing of the IP-SS hard segments results in non-crystallinity.

To summarize, IP-SS is considered the "hard segment" with respect to the chain structure, but the non-crystalline structure of IP-SS is certainly not the "hard domain." Therefore, we only use the term "hard domain" for the commercial TPU (i.e., Es-MD) as follows.

- (Original sentence, line 168) The FT-IR spectrum of **Es-MD** shows an intense band at 1703 cm^{-1} that corresponds to H-bonded urethane groups in the hard domain (Supplementary Fig. 11)
- (Original sentence, line 277) **Es-MD** provides a clearer spot in the equatorial region of its 2D WAXS pattern and a small change in the 1D pattern compared to that prior to stretching. The hard domain is oriented along the stretching axis with minor crystallite growth.

Comment 2: Can the authors specify what holds the network together in C-IP-SS elastomer, and furnishes the memory of the initial state? Is it just the hydrogen bonding between carbamate groups and carbamate/carbonate groups? Why do the effects persist longer than the strain induced crystallization?

Response to Comment 2: We appreciate the reviewer's constructive comment. It is proposed that two types of H-bonds hold the elastomer structure and furnish the memory of the initial state.

First, we emphasize that although the hard segments of C-IP-SS do not form a crystal structure, H-bonds exist between hard segments. The FT-IR spectrum of C-IP-SS presents H-bonds between hard segments at 1692 cm^{-1} (Fig. 4a in the revised manuscript). In this regard, the hard segments distinctly develop H-bonds with each other in the amorphous phase of C-IP-SS.

Second, hard-soft H-bonding is another constituent that holds the internal network of C-IP-SS. The commercial TPU only has non-H-bonded carbonyl groups and H-bonds between hard segments owing to the presence of the microphase separation (Supplementary Figure 11). In contrast, the FT-IR spectrum of C-IP-SS clearly presents the H-bonded carbonyl band between hard and soft segments newly appearing at 1718 cm^{-1} .

Fig. 4. FT-IR analysis of C-IP-SS in the static and dynamic states. a, The carbonyl region of the FT-IR spectrum with peak deconvolution for C-IP-SS. **b,** 2D gradient map of FT-IR

spectra with respect to stretching percentage. The red contour lines represent positive values of dA/dE , i.e., an increasing trend in absorbance as function of the degree of extension, and vice versa for the gray contour lines. **c**, Schematic illustration of the mechano-responsive changes in chain alignment and the associated H-bond arrays for **C-IP-SS**.

Supplementary Figure 11. The carbonyl region of FT-IR spectrum with peak deconvolution for Es-MD.

Comment 3: In line 92, the author mentioned the M_n of both TPUs are 1kDa, but the SI shows the M_n to be 16.5kDa and 30.4kDa.

Response to Comment 3: We apologize for the confusion. The information on line 95 shows the M_n values of macrodiols, not the synthesized TPUs. The sentence has been revised as follows:

- Line 95-97, "The number-average molecular weights (M_n s) of the carbonate- and ether-type macrodiols are 1 kg mol⁻¹ and their compositions in each self-healing TPU are almost equal (15 wt%)".

Comment 4: The authors should explain more about how they assign the peaks of IR for the C-IP-SS.

Response to Comment 4: According to the reviewer's advice, we have updated a more detailed explanation for the peak assignment in the revised manuscript. The commercial Es-MD exhibited two distinct peaks in the carbonyl region due to the clear microphase separation, whereas C-IP-SS exhibited four different peaks associated with carbonyl groups. The two typical bands for free (non-H-bonded) and H-bonded carbonyl groups were also observed in the spectrum of C-IP-SS at 1741 and 1692 cm^{-1} , respectively. The carbonyl band at 1718 cm^{-1} is reported to appear in the FT-IR spectrum of the carbonate-type TPUs [*Eur. Polym. J.* **47**, 959-972 (2011)]. This band is ascribed to the H-bonded carbonyl groups between the hard and soft segments. The carbonyl groups corresponding to the 4th band at 1645 cm^{-1} are related to the presence of neighboring aromatic disulfide (SS) units [*ACS Appl. Polym. Mater.* **2**, 285-294 (2020)]. The carbonyl groups produced the shoulder IR peak at lower frequencies when they were adjacent to the aromatic SS moiety because of steric hindrance and rotation restraint by aromatic rings.

- (Line 171-176), "On the contrary, the FT-IR spectrum of C-IP-SS shows four bands associated with carbonyl (C=O) groups (Fig. 4a); free (non-H-bonded) carbonyl groups of hard and soft segments (1741 cm^{-1}), H-bonded carbonyl groups between hard and soft segments (1718 cm^{-1}), and H-bonded carbonyl groups between hard segments (1692 cm^{-1})^{55,56}. The shoulder peak at a lower frequency (1645 cm^{-1}) corresponds to sterically hindered ester carbonyl groups that neighbor SS units⁵⁷."
- (Updated reference) 57. Zhou, J., Yang, Y., Qin, R., Xu, M., Sheng, Y. & Lu, X. Robust poly(urethane-amide) protective film with fast self-healing at room temperature. *ACS Appl. Polym. Mater.* **2**, 285-294 (2020)

Comment 5: From line 162 to 165, there are numerous grammatical mistakes.

Response to Comment 5: As the reviewer suggested, we have corrected the grammatical

errors in the description. We revised the sentence in lines 162–165 as follows.

- Line 171-176 in the revised manuscript, "On the contrary, the FT-IR spectrum of C-IP-SS shows four bands associated with carbonyl (C=O) groups (Fig. 4a); free (non-H-bonded) carbonyl groups of hard and soft segments (1741 cm^{-1}), H-bonded carbonyl groups between hard and soft segments (1718 cm^{-1}), and H-bonded carbonyl groups between hard segments (1692 cm^{-1})^{55,56}. The shoulder peak at a lower frequency (1645 cm^{-1}) corresponds to sterically hindered ester carbonyl groups that neighbor SS units⁵⁷."

Comments 6 & 7: In Figure 3b, why does the peak at 1692 cm^{-1} increase at low strain? And, in Figure 3b, why does the peak at 1650 cm^{-1} increase at high strain?

Response to Comments 6 and 7: In the revised Fig. 4b, the 2D gradient map is a useful tool for visualizing the band shift over a wide spectral range with respect to a perturbation change (the extent of stretching in the current study). However, it is limited to inform the sequence of the “relative spectral change” under perturbation. To study the effect of H-bonding on the mechano-responsive behavior, we focused on the stretching-dependent overall trend of intensity in the 2D map. In response to comment 8, we discuss the detailed change sequence of individual bands using 1D FT-IR spectra.

We note two trend changes in the 2D gradient map. First, the intensity of H-bonded carbonyl groups at 1718 , 1692 , and 1645 cm^{-1} increased with the degree of stretching (red contour), while the intensity of H-bond-free or loosely interacting carbonyl groups decreased (black contour). This suggests that drawing enhances the degree of H-bond ordering. Second, paying close attention to the contour of the H-bonded carbonyl area, the most significant change is observed in the band area for H-bonded carbonyl groups between hard and soft segments (1710 – 1730 cm^{-1}). This suggests that H-bonds between hard and soft segments play an important role in reversible stress-induced crystallization.

Fig. 4. FT-IR analysis of C-IP-SS in the static and dynamic states. **a**, The carbonyl region of the FT-IR spectrum with peak deconvolution for C-IP-SS. **b**, 2D gradient map of FT-IR spectra with respect to stretching percentage. The red contour lines represent positive values of dA/dE , i.e., an increasing trend in absorbance as function of the degree of extension, and vice versa for the gray contour lines. **c**, Schematic illustration of the mechano-responsive changes in chain alignment and the associated H-bond arrays for C-IP-SS.

Comment 8: In Figure 3b, it looks like that the peaks at 1718, 1692, 1645 cm^{-1} all increase at high strain, and the peak at 1741 cm^{-1} decreases. It looks like the H-bonded C=O groups increase in number as strain increases. However, the amount of hydrogen bond donors limits the number of hydrogen bonds in the elastomer. Are those hydrogen bond donors just partially bonded at unstrained state? Can the authors show how the exact IR absorption changes instead of the derivative? If we look at Figure 5c, the modulus decreases significantly from 0 to 40 degree. It seems that the hydrogen bonding is the main component that holds the network together. So, at 10 degree, I would expect that most of hydrogen donors are bonded. Then why do the authors observe an increase in bonded C=O groups when stretching the elastomer?

Response to Comment 8: As the reviewer mentioned, the number of H-bonding acceptors (carbonyl groups) is unchanged and independent of the degree of stretching. As the specimen is stretched, free or loosely H-bonded carbonyl groups are converted into H-bonded (ordered) groups through the formation of further H-bonding between the elongated polymeric chains.

It is possible because H-bond is “directional” [refer to Figure R2 and the book (Intermolecular and Surface Forces 3rd edition written by J. N. Israelachvili)]. In the amorphous phase, H-bond donors are poorly oriented, where H-bonds can be strongly formed in the optimal direction. If H-bonds are formed in the inappropriate direction, the interaction energy would be quite low. Extending polymer chains drives stress-induced conformational changes at the molecular scale, which orient H-bond donors/acceptors to the optimal direction. As a result, stress-induced crystallization is achieved.

154 INTERMOLECULAR AND SURFACE FORCES

FIGURE 8.3 Orientation-dependence of different types of bonds (schematic). Covalent bonds are the most “directional” with binding energies $w(\sigma)$ of order $\sim 100 kT$ at room temperature. Hydrogen bonds are less directional (varying up to $\sim 20^\circ$) and have lower energies ($\sim 10 kT$), while van der Waals bonds are not directional or only weakly directional and have the lowest energy of all ($\sim 1 kT$). Note, however, that there is no simple relationship between the strength and directionality of bonds; for example, ionic bonds are very strong and yet totally nondirectional.

Figure R2. Intermolecular and Surface Forces 3rd edition written by J. N. Israelachvili

In adhesive science, similar phenomena are observed. The adhesion force of the hydroxyl group-abundant adhesive can be measured at the molecular level or bulk scale. The adhesion force increased with pressing (stress) time at both measurement scales [Figure R3 in *J. Mater. Chem. A* **7**, 21944-21952 (2019)], although most hydrogen donors are bonded at the beginning of pressing. The pressing force changes the chain conformation, which adjusts the direction of the hydrogen donors. Thus, it takes time for molecules to become oriented in the optimal direction.

Figure R3. [*J. Mater. Chem. A* **7**, 21944-21952 (2019)]

In addition, we have shown the stretching-induced spectral change of 1D FT-IR spectra in response to the reviewer's suggestion. There are distinct trends in the spectral changes in the magnified figures (Figure R4). Along with the discussion of the 2D gradient map in the manuscript, the spectral absorbance band at more than 1740 cm^{-1} is weakened by stretching (blue arrow), but the spectral band at less than 1740 cm^{-1} is strengthened by stretching (red arrows). This corresponds to the red shift of the carbonyl band (*Polym. Degrad. Stabil.* **97**, 1794-1800 (2012)).

Figure R4. (a) The stretching-dependent change in the carbonyl region of the FT-IR spectra of C-IP-SS. Magnified spectra in the range of (b) $1760\text{-}1700\text{ cm}^{-1}$ and $1710\text{-}1630\text{ cm}^{-1}$.

Comment 9: Also, it is important to be sure that the change in sample geometry during straining does not influence any of the changes in intensity that are recorded.

Response to Comment 9: In the FT-IR measurement of the stretched specimens, the attenuated total reflection (ATR) mode was utilized. It is well-known that during the reflection measurement, the penetration depth of infrared light into the sample is typically 0.5–2 μm . In other words, the light during the reflection mode is concentrated at a localized spot on the specimen surface. Therefore, the change in sample geometry (length, width, and thickness) has an insignificant effect on the FT-IR spectrum.

To further ensure the reliability of spectral results, the measurement spot was fixed to the center of the dumbbell-shaped specimens whether stretched or not, and the FT-IR measurement for each sample was repeated five times.

Comment 10: In Figure 3c, I do not think there is a strong interaction between carbonate groups, at least not that is comparable in strength to the hydrogen bonding. The authors put this interaction together with other hydrogen bonding, which is confusing. This requires either a reference or stronger independent justification.

Response to Comment 10: We appreciate the reviewer’s critical comment. We agree with the reviewer's opinion that the carbonate–carbonate interactions are much weaker than other H-bonds. Therefore, we have subtracted the illustration of carbonate–carbonate bonds from Fig. 3c in the revised manuscript. The revised figure is shown below:

- Revised Fig. 3. (Fig. 4 in the revised manuscript).

Fig. 4. FT-IR analysis of C-IP-SS in the static and dynamic states. a, The carbonyl

region of the FT-IR spectrum of C-IP-SS with peak deconvolution. b, 2D FT-IR gradient map with respect to stretching percentage. The red contour lines represent positive values of dA/dE ; i.e., an increasing trend in absorbance as a function of the degree of extension, and vice versa, for the gray contour lines. c, Schematic of the mechano-responsive changes in chain alignment and the associated H-bond arrays for C-IP-SS.

Comment 11: *When calculating the energy required to form the crystallite, have the authors considered that the crosslinking density is also changing? And the material is not identical at temperatures above T_g .*

Response to Comment 11: In the current study, the TPUs do not contain chemical crosslinking. The rubber-like elasticity and mechanical strength of TPUs originate from non-covalent interactions such as pi-pi stacking and/or hydrogen bonding. Therefore, we did not consider the crosslinking density of C-IP-SS and its change upon stretching.

We agree with the reviewer's comments. We studied the point "the material is not identical at temperatures above T_g " in the manuscript. As shown in the stress-strain (S-S) curves at various temperatures and force versus (vs.) temperature plots in Fig. 6 (revised version), the S-S curve trend and the slope on the force vs. temperature plots exhibited critical changes above T_g (~ 0 °C). This is because the mechanical stretching produced different internal structures and related energy states (more stable and ordered crystals (irreversible deformation) below T_g but metastable crystals (reversible deformation) above T_g). Hence, we calculated the internal energy required to form the strain-induced crystallization using only the data above T_g .

- Original sentences (Line 308-317): The steeper slope indicates that a higher internal-energy change results in more-stable and ordered crystals, which is associated with the following observations. C-IP-SS was drawn and released above its T_g ; the original strain is then recovered because of its relatively unstable crystal phases. However, C-IP-SS is not reversible below T_g because relatively stable crystals are formed. The energy (ΔU) required to form the stress-induced metastable crystal is calculated to ~ 6.47 cal cm^{-3} above T_g , the details of which are described in

Supplementary Note 1 (Supplementary Figs. 21-23 and Supplementary Table 4). At a TPU density of 1.25 g cm^{-3} , this value can be converted into $\sim 5.17 \text{ cal g}^{-1}$. The calculated heat of the metastable crystal is 3–10-times lower than those ($15\text{--}65 \text{ cal g}^{-1}$) of stable commercial-polymer crystals⁶⁷, which qualitatively shows that the instantly formed strain-induced crystal is unstable.

[Response to Reviewer #3]

Overall comment: *This manuscript reports a novel thermoplastic polyurethane elastomer that displays an improved combination of strength and self-healing capacity. The strength of the manuscript relates to the mechanistic part where the fundamental aspects of the reversible strain-induced crystallization are unraveled. This part of the publication is designed, performed and interpreted at high scientific level. Nevertheless, the novelty and quality of the biological evaluation is rather limited.*

Response to Comment: We are very grateful for the reviewer's evaluation of our work as being of "high scientific level." In order to compensate for the lack of biological studies and improve the entire manuscript, we have addressed all issues in the reviewer's report.

Comment 1-1: *As compared to previous work on self-healing polyurethanes (e.g. reference 47 by the same group) the improvement of the tensile strength should be regarded as incremental optimization rather than a breakthrough development that will create entirely new avenues of research.*

Response to Comment 1-1: This study includes the previously reported self-healing TPU (reference 47) = E-IP-SS and compares the mechanical properties of C-IP-SS and E-IP-SS. C-IP-SS gives the quantum jump in mechanical performance through a mechano-responsive dual mode (healing in static mode and toughening in dynamic mode). This concept is based on a reversible disorder-to-order transition of the H-bonding array upon straining, which is the inherent characteristic of the carbonate-type TPU. In contrast, the ether-type self-healing TPU (E-IP-SS) exhibited little phase transition during stretching because the ether-type soft segment was not involved in the H-bonding. Although the change in the soft segment structure is a simple tuning, this molecular approach renders the inherent property (hard-soft H-bonding and mechano-responsive behavior) and results in superior strength. Therefore, this molecular concept is a new strategy compared to our previous report, not just optimization of the material preparation.

Comment 1-2: *Moreover, the enhanced strength goes at the expense of ductility. Unfortunately, the authors only discuss the positive outcome of this study (improved strength*

and toughness) without properly discussing the negative results (reduced ductility, as shown in Figure 1c).

Response to Comment 1-2:

As the reviewer pointed out, it is a fact that the ductility of C-IP-SS is lower than that of E-IP-SS. Therefore, we added an explanation for the reduced ductility of C-IP-SS in the revised manuscript.

- Line 106-108. "Although C-IP-SS has the lowest M_w among the TPUs, its UTS of 43 MPa is higher than the other two control TPUs. It exhibits two-fold lower extensibility (450%) but a 2.8-fold greater toughness (75 MJ m^{-3}) than E-IP-SS (Fig. 1a, c)."

From the perspective of tensile properties, the improved tensile strength pays a significant penalty on the extensibility in most cases of polymeric materials, including self-healing TPUs because the strength and ductility are exclusive properties. The mechanical strength requires high chain rigidity or high crystallinity, which inevitably causes the deterioration of ductility due to the restricted chain flow and deformation.

In case of self-healing materials, the ductile-to-stiff transition significantly degrades the self-healability, i.e., the trade-off between strength and self-healing. In this regard, the novelty of the current study is definitely highlighted; the self-healing ability deteriorates at ambient temperature ($\sim 35 \text{ }^\circ\text{C}$) even after ductile-to-stiff tuning. Therefore, breaking the trade-off relation between mechanical and healing performances is an influential advance in self-healing research fields.

Comment 2: *Regarding the biocompatibility studies it should be stressed that the authors do not properly describe the intended application (i.e. muscle tissue replacement) in the introduction. They introduce the intended application only at the end of the manuscript (p17) by defining the concept of "precursor material for artificial muscle", but this concept is not defined adequately from a biological/clinical perspective. What is a "precursor material for artificial muscle", and why would the newly developed TPU be a suitable candidate biomaterial? Why are the improved strength and toughness vs. reduced ductility of the newly developed TPU relevant for muscle tissue replacement?*

Response to Comment 2: We appreciate the reviewer’s critical comment. We lacked understanding of the artificial muscles. Thus, we have removed the content relating to muscle tissue replacement.

This TPU is regarded as a suitable structural material for healthcare e-skins and soft robots due to a relevant combination of its appropriate stiffness, high mechanical performance, and great damage-recovery. Flexible and stretchable structure materials are essential for such applications. They have the following requirements: a similar Young’s modulus of 10^4 – 10^7 Pa to those of soft tissues, mechanical robustness, self-healing, and biocompatibility. Elastomers such as polydimethylsiloxane (PDMS) and hydrogels are typically used as the flexible material in these devices. However, current elastomers and hydrogels lack self-healing capability, which is among the distinctive characteristics that mimic natural organisms. Please refer to our previous report (Figure R5; ref. 13).

Figure R5. Soft self-healing materials for E-skin. Ref. 13

- 13. Shin, S.-H. et al. Ion-conductive self-healing hydrogels based on an interpenetrating polymer network for a multimodal sensor. *Chem. Eng. J.* **371**, 452-460 (2019).

We have updated the contents as follows.

- (Revised version, Line 319-326) E-skin and wearable soft robots for healthcare

mimic the nature of human body organs. Flexible and stretchable structural materials are essential for such purposes and have many requirements¹³, including similar Young's moduli to those of soft tissue (10^4 – 10^7 Pa), mechanical robustness, self-healing, and biocompatibility. However, current elastomers and hydrogels lack self-healing capabilities, which is a distinctive characteristic required to mimic natural organs. Therefore, this TPU is regarded as a suitable structural material for healthcare e-skin and soft robots due to the relevant combination of its exploitable stiffness, high mechanical performance, and damage-recovery.

Comment 3: *The in vivo implantation study itself is qualitative at best and does not provide sufficient evidence to justify the authors' claims on superior biocompatibility and reduced inflammatory responses. The authors should provide more quantitative evidence (e.g. counting of inflammatory cells or staining of other inflammatory markers) to justify these claims. The selected implantation model does not allow to draw conclusions on the functionality of the implanted TPU, while the authors should have selected an intramuscular instead of subcutaneous implantation site to confirm the suitability of the developed biomaterial for muscular replacement. The authors should consider to compare the biological performance of their novel C-IP-SS polyurethane with conventional non-self healing polyurethanes (such as E-IP-SS and Es-MD shown in Figure 1) to confirm the superiority of self-healing polyurethanes over conventional polyurethanes. To provide further evidence for the cytocompatibility of the newly developed materials, in vitro cell culture studies should be performed as well.*

Response to Comment 3: As the reviewer suggested, we conducted *in vitro* cytotoxicity tests with four different types of cells (mouse macrophage normal RAW 264.7, human breast carcinoma MDA-MB-231, human epidermoid carcinoma KB, and Chinese hamster ovary normal CHO-K1). The cytotoxicity of C-IP-SS was compared to that of commercial TPU (Es-MD). The *in vitro* cell viability test included negative control (pristine culture medium), positive control (culture medium with 5% DMSO), C-IP-SS extract-containing medium, and Es-MD extract-containing medium. The results have been added to the manuscript and

Supplementary Figure 24. We have updated the reference in the manuscript as ref. number 68 to support the *in vitro* cytotoxicity test.

- (Line 327-340) As the last requirement, the physiological adaptability of **C-IP-SS** was evaluated by examining the *in vitro* cytotoxicities of **C-IP-SS** and commercial **Es-MD** in four types of cell, namely human breast carcinoma MDA-MB-231, human epidermoid carcinoma KB, Chinese hamster ovary normal CHO-K1, and mouse macrophage normal RAW 264.7, following the ISO 10993-5 guidelines (Supplementary Fig. 24)⁶⁸. Cytotoxicity testing began with liquid extracts of the plastic materials. **C-IP-SS** was immersed in a cell growth medium with the plastic sample (area: 1 cm², thickness: 2 mm) in 1 mL of the medium at 36.5 °C for 72 h. The four types of cell were cultured in a complete growth medium, after which the culture medium was replaced with (1) neat cell culture medium (negative control), (2) cell culture medium with 20% (v/v) **C-IP-SS** extract (experimental group), or (3) cell culture medium with 5% dimethyl sulfoxide (DMSO) (positive control). The cells were then incubated for an additional 24 h and cell viabilities were evaluated using a colorimetric assay. **Es-MD** was tested using the same procedure. Both **C-IP-SS** and **Es-MD** exhibited consistently negligible cytotoxicity to the four types of mammalian cells; i.e., more than 80% of the cells were viable compared to the negative control^{68,69}.

- (Methods section, Line 496-514) *In vitro* cytotoxicity testing was performed according to the ISO 10993-5 international standard⁶⁹. Cytotoxicity experiments began with liquid extracts of the plastic materials. A 1 cm² **C-IP-SS** film sample sterilized with 70% aqueous ethanol solution was immersed in 1 mL of cell growth medium at 36.5 °C for 72 h. The culture medium extract was filtered through a syringe filter. Human breast carcinoma MDA-MB-231 cells, human epidermoid carcinoma KB cells, normal Chinese hamster ovary CHO-K1 cells, and normal mouse macrophage RAW 264.7 cells were purchased from the Korea Cell Line Bank and cultured in a Dulbecco's modified Eagle's medium (DMEM) supplemented with 10% fetal bovine serum, 100 U mL⁻¹ penicillin G, 100 µg mL⁻¹ streptomycin, and 0.025 µg mL⁻¹ amphotericin B at 36.5 °C for 24 h in 5% CO₂ atmosphere. During *in vitro* cytotoxicity testing, the culture medium was replaced with neat culture medium (negative control), culture medium with 20% (v/v) **C-IP-SS** extract, or culture

medium with 5% DMSO (positive control). The cells were then incubated for an additional 24 h to expose the cell to the extract. The CCK-8 cell counting kit (Dojindo, Inc., Rockville, MD, USA) was used to evaluate cell viability. Incubated media were transferred to fresh 96-well plates for colorimetric assessment at 450 nm using a microplate reader. **Es-MD** cytotoxicity testing followed the same procedure. An absorbance intensity that corresponds to less than 80% cell viability compared to the negative control indicates that the sample is not biocompatible. Data from quintuplicate samples are expressed as means \pm standard deviations. All data were evaluated using a Student's t-test at a not significance (NS) of $p > 0.05$ and a significance of $p < 0.01$ (**).

- 68. Loh, X. J., Sng, J. B. C. & Li, J. Synthesis and water-swelling of thermo-responsive poly(ester urethane)s containing poly(ϵ -caprolactone), poly(ethylene glycol) and poly(propylene glycol). *Biomaterials* **29**, 3185-3194 (2008).

Supplementary Figure 24. *In vitro* cell viability tests of negative control (pristine cell culture medium), positive control (cell culture medium with 5% DMSO), **C-IP-SS** extracts, and **Es-MD** extracts using four different types of cells: (a) Chinese hamster ovary normal CHO-K1, (b) human breast carcinoma MDA-MB-231, (c) human epidermoid carcinoma KB, and (d) mouse macrophage normal RAW 264.7.

Comment 4: Finally, statistical analysis is required for the presented mechanical and biological tests in Figures 1, 2, 4 and 6, but this analysis is lacking unfortunately. Consequently, reproducibility of the results is currently questionable.

Response to Comment 4: We agree with the reviewer’s comment. Our data presentation presented misleading results. We updated the statistical analysis as follows.

Tensile tests for Figure 1 & 2

We mistakenly did not fill in the number of tensile measurements. We have added the information in the Methods section and Supplementary Table 2. Figs. 1 and 2 show the representative curves of each sample. The Supplementary Information includes the mean \pm standard deviation of quintuplicate samples.

- (Supplementary Information) Supplementary Table 2. Information on tensile properties of TPU films and the degree of recovery after self-healing. The data of quintuplicate samples are expressed as mean \pm the standard deviation.

Entry		C-IP-SS ^a	E-IP-SS ^b	Es-MD
Virgin sample	Young's modulus (MPa)	15.5 \pm 0.8	1.45 \pm 0.1	8.86 \pm 0.4
	UTS (MPa)	42.9 \pm 1.4	6.76 \pm 0.4	35.8 \pm 0.8
	Elongation at break (%)	480 \pm 3	920 \pm 43	880 \pm 8
	Toughness (MJ m ⁻³)	75.1 \pm 2.3	26.9 \pm 2.7	115 \pm 2
Cut & healed sample	Young's modulus (MPa)	15.3 \pm 0.4	1.47 \pm 0.1	- ^c
	UTS (MPa)	33.1 \pm 0.4	5.96 \pm 0.1	-
	Recovery of UTS (%)	77.2	88.2	-
	Elongation at break (%)	400 \pm 6	920 \pm 11	-
	Recovery of elongation (%)	82.7	99.6	-
	Toughness (MJ m ⁻³)	48.4 \pm 0.6	20.6 \pm 0.2	-
	Recovery of toughness (%)	64.4	76.6	-

^aSelf-healing for 48 h at 35 °C. ^bSelf-healing for 2 h at 25 °C. Tensile data obtained from the previous report (*Adv. Mater.* **2018**, *30*, 1705145). ^cNo mechanical recovery was observed.

(Method section; Line 451-452): The tensile test data of quintuplicate samples are expressed as mean \pm the standard deviation.

Rheology test for Fig. 4 (revised Fig. 5)

Every single data point of rheological analysis is the mean of five measurements. The instrument automatically calculates it and gives the resultant values. Since rheological results are logarithmic values, they are insensitive to deviation. Therefore, in general, as shown in Figure R6 and R7, rheological data are expressed as points without standard deviation.

Figure R6. *Chem. Soc. Rev.* **39**, 3528-3540 (2010).

Figure R7. *Macromolecules* **40**, 3378-3387 (2007).

In vivo experiments for Fig. 6 (the revised Fig. 7)

The in vivo test results were obtained from the samples implanted in male Sprague–Dawley rats (8-weeks-old, 250-300 g) (n = 4). We have updated the precise details of the results of the in vivo biological tests in the Supplementary Information. Semi-quantitative histocompatibility evaluation was conducted by a pathologist in a contract clinical research organization regarding ISO 10993-6.

Supplementary Table 5. Average score of local irritation reaction to subcutaneous implants from histological data (n = 4)

Cell type/ tissue response	Experimental (C-IP-SS)	Control	
		Negative (HDPE)	Positive (DMSO-TPU)
Inflammation			
polymorphonuclear cells ^a	0.75	0.29	3.00
Lymphocytes ^a	0.25	0.14	2.50
Plasma cells ^a	0.00	0.00	0.00
Macrophages ^a	0.00	0.00	0.25
Giant cells ^a	0.00	0.00	0.00
Necrosis ^b	0.00	0.00	2.25
A: subtotal			
inflammation score (×2)	2.00	0.86	16.00
Neovascularization^c			
Fibrosis ^d	1.00	1.29	2.75
Fatty infiltrate ^e	0.00	0.00	0.25
B: tissue response subtotal			
	1.00	1.86	5.25
Total (A+B)	3.00	2.72	21.25

Score (= experimental – negative control) ^f	0.29	-	18.53
Traumatic necrosis	0	0	0
Foreign debris	0	0	0
No. sites examined	8	8	8

^a Mean histopathological score of reaction based on the number of cells per high-powered field at 400x: 0 = 0 cells; 1= 1-5 cells; 2 = 5-10 cells; 3 = heavy infiltrate; 4 = packed.

^b 0 = None; 1 = minimal; 2= mild; 3 = moderate; 4 = severe.

^c 0 = None; 1 = minimal capillary proliferation, focal, 1–3 buds; 2= groups of 4–7 capillaries with supporting fibroblastic structures; 3 = broad band of capillaries with supporting structures; 4 = extensive band of capillaries with supporting fibroblastic structures.

^d 0 = None; 1 = narrow band; 2= moderately thick band; 3 = thick band; 4 = extensive band.

^e 0 = None; 1 = minimal amount of fat associated with fibrosis; 2= several layers of fat and fibrosis; 3 = elongated and broad accumulation of fat cells about the implant site; 4 = extensive fat completely surrounding the implant.

^f Non-irritant (0.0 up to 2.9) < slight (3.0 up to 8.9) < moderate (9.0 up to 15.0) < severe (more than 15.0).

The local irritation effects were evaluated by a comparison of the tissue response caused by the tested implant to that caused by the negative control. The stained tissue cross-sections were examined by a pathologist in a public contract clinical research organization using an optical microscopy at high-powered field of 400x, and the pathologist scored local irritation responses based on the number of cells and the tissue change. The all scores are average values (n = 4). The subtotal inflammation score was multiplied by 2 and added to the subtotal score of neovascularization, fibrosis, and fatty infiltrate. Then, the value was subtracted by the score of negative control. According to the final score, the sample can be considered as non-irritant (0.0 up to 2.9), slight (3.0 up to 8.9), moderate (9.0 up to 15.0), and severe (more than 15.0).

In vitro experiments for Supplementary Figure 24

Please refer to Response to Comment 3 of Reviewer #3.

We conducted a statistical analysis for in vitro experiments. The sample data of quintuplicate samples are expressed as mean \pm standard deviation. All data were evaluated using a Student's t-test at a non-significance (NS) of $p > 0.05$ and a significance of $p < 0.01$ (**).

Supplementary Figure 24. *In vitro* cell viability tests of negative control (pristine cell culture medium), positive control (cell culture medium with 5% DMSO), **C-IP-SS** extracts, and **Es-MD** extracts using four different types of cells: (a) Chinese hamster ovary normal CHO-K1, (b) human breast carcinoma MDA-MB-231, (c) human epidermoid carcinoma KB, and (d) mouse macrophage normal RAW 264.7. The data of quintuplicate samples are expressed as mean \pm the standard deviation. All data were evaluated using a Student's t-test at a not significance (NS) of $p > 0.05$ and a significance of $p < 0.01$ (**).

Raw data provision

We provided raw data to help improve reviewer confidence in our data. Please find the raw data in the link provided (<https://figshare.com/s/9f047f83055eab32280b>).

Comment 5: Figure caption 6: "*in vivo* cytotoxicity" should be "*in vivo* histocompatibility" since the figure shows tissue response rather than cellular response.

Response to Comment 5: This was a misuse of the term. According to your advice, the "cytotoxicity" was replaced by "histocompatibility" in the revised manuscript.

- **Fig. 7. Biocompatibility test of C-IP-SS.** **a,** Experimental procedure for *in vivo* histocompatibility testing: rat subcutaneous connective tissues with negative control (NC, HDPE), positive control (PC, DMSO-containing polyurethane), and the C-IP-SS film. **b,** Representative histopathologic tissue images after 12 weeks of healing. Arrows indicate inflammatory cells (red: polymorphonuclear cells; black: lymphocytes).

Comment 6: Details on sterilization of implanted materials are lacking.

Response to Comment 6: We appreciate the reviewer's comment. The disk-shaped TPU specimens were sterilized using ethylene oxide gas for 12 h prior to implantation. The information on the sterilization procedure has been added in the revised manuscript.

- Line 348-349, "For sterilization, they were sealed in packs and exposed to ethylene oxide gas for 12 h."

REVIEWER COMMENTS

Reviewer #1 (Remarks to the Author):

The authors have addressed all of my concerns.
And i believe the paper can become published now

Reviewer #2 (Remarks to the Author):

Based on the responses, the authors have addressed most of my concerns, except comment 8 and comment 11, but these are the most important ones. My overview of this manuscript is unchanged: I worry that much of the analysis remains somewhat speculative and that some aspects of the mechanism are unknown, and the impact of the paper truly lies with just how remarkable the reported properties are. The argument for the novelty of the properties seems potentially to justify its publication in NCOMMS, but I am not an expert in this and am unsure of how to evaluate it. I will defer to other reviewers on this point. I am most concerned with the mechanistic insights, and there I still have questions that would have to met prior to acceptance as follows.

After reading the authors' responses on comment 8, I do not think the authors have answered my question, and I am still confused about the IR change with respect to the stretch. It seems like the authors also agree that most of the hydrogen donors are bonded at the unstrained state, then why do we see a decrease of unbonded C=O group at 1741 cm^{-1} while stretch increase? This means more hydrogen bonds are forming while the material is stretched. Considering the number of hydrogen donors are constant, and most of them are already bonded, what is the origin of these extra hydrogen bonds? The authors said, "In the amorphous phase, H-bond donors are poorly oriented, where H-bonds can be strongly formed in the optimal direction." Then, they also said "Extending polymer chains drives stress-induced conformational changes at the molecular scale, which orient H-bond donors/acceptors to the optimal direction." So in both situations, the H-bonds are in the optimal direction? I am confused. In the adhesion example that the authors mentioned, it seems like they are trying to state that H-bonds take time to reorient to the optimal direction during polymer extension, but I don't think it explains the decrease of the unbonded C=O group during the extension. If H-bonds become more labile (or less oriented to optimal direction) during chain extension, we should observe the peak at 1741 cm^{-1} increase instead of decrease, since there should be more unbonded C=O bonds.

In particular, in figure R4, the absolute changes in each peak are tiny, and I am not sure that the conclusions are justified. The isosbestic point looks weird to me. It looks like the whole curve is just redshifted a little bit. I have a hard time placing as much significance on the data as the paper requires.

For the response of comment 11, the decrease of energetic force versus temperature is not only caused by crystalline formation but also caused by the relaxation of chains before the crystalline formation (low strain such as 20% and 100%), since H-bonding becomes more labile as temperature increases. I don't think the calculation of crystallization is correct because the internal energy change is not only due to crystallization.

Reviewer #3 (Remarks to the Author):

The comments of all reviewers have been adequately addressed by the authors.

Point-to-point responses to referees

Contents

Responses to reviewer #1: Page 2

Responses to reviewer #2: Page 3

Responses to reviewer #3: Page 10

Reviewers' comments are in italic

Responses are highlighted in gray

The updated contents are highlighted in yellow

The dragged contents from the original manuscript are in sky-blue

The materials only for revision are in purple

[Response to Reviewer #1]

***Overall comment:** The authors have addressed all of my concerns. In addition, I believe the paper can be published now.*

Response: We appreciate your positive evaluation of our manuscript.

[Response to Reviewer #2]

Overall comment: *Based on the responses, the authors have addressed most of my concerns, except comments 8 and 11, but these are the most important ones. My overview of this manuscript is unchanged: I worry that much of the analysis remains somewhat speculative and that some aspects of the mechanism are unknown, and the impact of the paper truly lies with just how remarkable the reported properties are. The argument for the novelty of the properties seems to justify its publication in NCOMMS, but I am not an expert in this and am unsure of how to evaluate it. I will defer to other reviewers on this point. I am most concerned with the mechanistic insights, and there I still have questions that would have to meet prior to acceptance as follows.*

Response: We sincerely appreciate your valuable comment. We would like to further clarify the explanation for the mechano responsive H-bond array concept to you. We have, thus, addressed the issues in the comments below with more detailed and convincing interpretations of H-bonding upon straining. We hope that you find our responses sufficient and satisfying.

Comment 1: *After reading the authors' responses on comment 8, I do not think the authors have answered my question, and I am still confused about the IR change with respect to the stretch. It seems like the authors also agree that most of the hydrogen donors are bonded in the unstrained state, then why do we see a decrease of unbonded C=O group at 1741 cm^{-1} while stretch increase? This means that more hydrogen bonds are formed while the material is stretched. Considering that the number of hydrogen donors is constant, and most of them already bond, what is the origin of these extra hydrogen bonds? The authors have stated that in the amorphous phase, H-bond donors are poorly oriented, where H-bonds can be strongly formed in the optimal direction." Then, they also said "Extending polymer chains drives stress-induced conformational changes at the molecular scale, which orient H-bond donors/acceptors to the optimal direction." Therefore, in both situations, the H-bonds are in the optimal direction? I am confused. In the adhesion example mentioned by the authors, it seems that they are trying to state that H-bonds take time to reorient to the optimal direction during polymer extension, but I do not think it explains the decrease of the unbonded C=O*

group during the extension. If H-bonds become more labile (or less oriented in the optimal direction) during chain extension, the peak at 1741cm^{-1} increases instead of decreasing, since there should be more unbonded C=O bonds.

Response to Comment 1: We thank you for your comment. To interpret the stretching-dependent change in the FT-IR spectra more clearly, a schematic representation of the change in H-bond states and the relationship between the H-bond geometry and chain conformation has been added in the supplementary information as Supplementary Fig. 12. Upon straining, to be specific, the number of H-bond donors (and acceptors) remains constant, but the H-bond state and related strength are changed (Supplementary Fig. 12a). The formation of stronger H-bonds between stretched polymeric chains is because H-bond is "directional", as described in the previous response letter.

A second type of H-bond directionality is associated with the acceptor and measured by the H-A-R' angle, ϕ (2.1.VII). The effect is due to the uneven spatial localization of the lone-pair electron densities around the acceptor and can be observed only when this localization becomes significant as, for instance, around the sp^2 -hybridised oxygen of the carbonyl. An extended CSD study on conventional N-H...O and O-H...O bonds is available (Murray-Rust and Glusker, 1984). A beautiful example of lone-pair directionality (Fig. 2.2) has been obtained by Allen *et al.* (1997) by comparing the distribution of the angular values obtained from a CSD search on weak X-H...S=C< and conventional X-H...O=C< bonds (X=N and O). It shows that both types of bonds have a strong **directional** preference in the plane of the >C=S or >C=O groups with the only quantitative difference that the H...S-C angles are centered somewhat below 110° while the H...O-C ones gather around 120° , in good agreement with the common belief that lone pairs of S and O have greater p and sp^2 character, respectively.

Figure R1. H-bond directionality. Book: The Nature of the Hydrogen Bond: Outline of a Comprehensive Hydrogen Bond Theory By Gastone Gilli, Paola Gilli

This is consistent with the theoretical calculation for OH...O bonds using *ab initio* quantum mechanics on a model system.¹⁸ However, this does not imply that linearity is not the most stable configuration for a hydrogen bond between isolated donor and acceptor molecules or between hydrogen-bonded dimers in the gas phase. Linear hydrogen bonding between one donor and one acceptor is energetically optimal, but is not common. Bent hydrogen bonds are often found in bifurcated configurations (Figure 3.2); in configuration 1, one hydrogen atom is located between three electronegative atoms, covalently bound to one and hydrogen bonded to the other two. The bifurcated hydrogen bond was first found in the crystal structure of glycine.¹⁹ The bifurcation in configuration 2² is rarely observed in the crystalline state. To avoid confusion, the “three-centered” description was proposed for configuration 1 by Jeffery and Saenger.⁴ Configuration 2 is observed in nucleosides only in conjunction with configuration 3. Nevertheless, the strong angular dependence of hydrogen bonding exists and it is useful for constructing artificial molecular organizations having well-defined structures.

Figure R2. H-bond directionality. Book: *Supramolecular Design for Biological Applications* by Nobuhiko Yui. Please note the sentence “Linear hydrogen bonding between one donor and one acceptor is energetically optimal, but is not common”.

We would like to further clarify the reviewer’s question regarding H-bond directionality. The H-bond strength-directionality relationship is illustrated in Supplementary Fig. 12b and 12c. The directionality of H-bonds reveals that the bond strength is dominantly dependent on the bond angle; the closer the angle is to 180°, the stronger and more stable is the bond. Hence, the linear position with a bond angle of 180° is not always common (Supplementary Fig. 12b).

In this regard, the stretching-induced conformational change of polymeric chains can affect the H-bond state and corresponding strength because the bond angle of intermolecular H-bonds is instinctively changed (Supplementary Fig. 12c). In the unstrained state, the poor chain alignment with the cis-dominant conformation is less favorable for the formation of H-bonds due to the distorted bond angle. Upon straining, however, the evolution of the trans conformation with higher chain alignment provides the optimal position for linear H-bond formation. Therefore, the steps involved in the process of the stretching-dependent change FT-IR spectra is summarized as follows: 1) uniaxial stretching, 2) lateral chain alignment with conformational change, 3) enhanced linearity and strength of H-bonds, and finally, 4) FT-IR spectral change (the decrease of loosely H-bonded C=O and the increase of strongly H-bonded C=O).

Supplementary Figure 12. (a) Stretching-dependent change of H-bond states and related FT-IR bands of C-IP-SS with a constant number of H-bond donors (carbonyl groups), (b) energy distributions of H-bonds dependent on the bond angle (ϕ)²¹, (c) and the effect of chain conformational change upon straining on the linearity and strength of H-bond.

As described in the previous response letter, H-bond directionality is a well-known phenomenon in adhesive science. The angle (orientation) tendency of the hydrogen bond results in the time tendency of the H-bond interaction strength. The molecular interaction can be measured using a surface force apparatus (SFA). Hydrogen donors exist at the beginning of pressing, but they form weak interactions because they are not oriented in the optimal direction. Therefore, the molecular interaction increases with the pressing (stress) time. The pressing force changes the chain conformation, which adjusts the direction of the hydrogen donors. Thus, it takes time for molecules to become oriented in the optimal direction.

In other words, not all H-bond donors participate in linear H-bonds. H-bond donors need re-ordination and conformational changes to form H-bonds in the optimal direction. In many cases, external stress drives the re-ordination and conformational changes of H-bond donors. Please refer to the following three references:

formation of physical bonding (e.g., hydrogen bonding and hydrophobic interaction).^{12,24} The contact time effect could be explained by the fact that the increase of the contact time provides more time for chitosan chains to optimize their orientation, forming more hydrogen bonds and strengthening the hydrophobic

Figure R3. SFA experiments and the orientation dependency of hydrogen bonds. [Phys. Chem. Chem. Phys., 2019, 21, 20571-20581].

dependent adhesion tests (Fig. 3), we can conclude the following. First, bulk and surface molecular rearrangements are required to enable hydrogen bonding. The existence of a critical t_c indicates that the rearrangement/reorientation of polymer chains and molecular groups is a requisite for extensive interfacial hydrogen bonding. That the critical t_c increases with decreasing bulk and molecular mobility further supports this conclusion. Second, hydrogen-bond formation accelerates other attractive interactions. For the blocked catecholic semi-rigid polymer, F_{ad} increased only slightly ($\sim 230 \text{ mN m}^{-1}$ at $t_c = 3,600 \text{ s}$), whereas the adhesion of exposed catecholic polymer (Fig. 3d) increased from 70

Figure R4. SFA experiments and the orientation dependency of hydrogen bonds. [Nature Materials 2014, 13, 867–872].

Here, unlike in the previous SFA study, the effects of contact time on chitosan interactions in aqueous solutions were investigated using an SFA because longer contact times can induce molecular reorientations and interdigitations that can significantly increase adhesion and cohesion.²⁵ Our results suggest that the contact time plays a critical role in the strong cohesion between chitosan films, which is likely due to molecular reorientations and interdigitations of the chitosan film during prolonged contact times.

Figure R5. SFA experiments and the orientation dependency of hydrogen bonds. [Langmuir 2013, 29, 46, 14222–14229].

Comment 2: In particular, in figure R4, the absolute changes in each peak are tiny, and I am not sure that the conclusions are justified. The isosbestic point looks weird to me. It looks like the whole curve is redshifted slightly. I have a hard time placing as much significance on the data as the paper requires.

Response to Comment 2: We would like to thank you for your comment. As you pointed out, the stretching-induced FT-IR spectral change was relatively small (Figure R4 in the previous response letter). It would appear that the vibrational modes of individual molecules (in the polymeric chains) are indirectly affected by the mechanical stress and resultant conformational change. A similar trend (relatively slight change of FT-IR spectra upon straining) has been observed in other polymeric materials (*Macromolecules* **36**, 6114-6126 (2003); *ACS Appl. Mater. Interfaces* **11**, 13665-13675 (2019)). To ensure the reliability of the FT-IR data, the spectral results were obtained and processed through the following procedures: 1) multiple FT-IR scans and measurements to collect the reproducible spectra, 2) the necessary pre-treatment such as baseline correction and normalization to find a tendency, and 3) the 2D gradient map to clearly visualize the spectral change trend. Therefore, we are convinced that the FT-IR results are reliable and offer meaningful evidence.

In addition, the isosbestic point may not exist because the FT-IR curve was red-shifted with the stretching. However, As we have stated in the previous response, the purpose of the 2D gradient map was to trace the stretching-dependent overall trend of spectral change by visualizing the intensity increase or decrease at a specific wavenumber during "red shift" of the carbonyl band.

The description for the pre-treatment of FT-IR data has been added in the revised manuscript.

- (Line 460–462): To ensure the data reliability and subsequently draw the 2D gradient map, the essential pre-treatment procedures of FT-IR data, including the baseline correction and normalization, were performed.

Comment 3: For the response of comment 11, the decrease in energetic force versus temperature is not only caused by crystalline formation but also caused by the relaxation of

chains before the crystalline formation (low strain such as 20% and 100%), since H-bonding becomes more labile as temperature increases. I do not think the calculation of crystallization is correct because the internal energy change is not only due to crystallization.

Response to Comment 3: Thank you very much for your valuable comment. The objective of the thermodynamic calculation was to obtain the required internal energy not only for crystallization, but also during the stretching-induced phase conversion. In line 289 of the manuscript, the subtitle "Practical calculation of reversible mechano-responsive crystallization" was slightly misleading; hence, we have rephrased the term "crystallization" as "phase conversion".

Although the calculated internal energy required for the phase change was not only due to the crystallization, the crystallization contributed mostly to the internal energy change because it was the dominant phenomenon during the strain-induced phase conversion according to several experimental results. Of course, other phenomena, including chain relaxation, were also involved in the phase conversion, but their contribution to the internal energy seems to be negligible. Therefore, we believe that it is reasonable to consider that the calculated internal energy for the phase change is mainly ascribed to the strain-induced crystallization.

[Response to Reviewer #3]

Overall comment: The comments of all reviewers have been adequately addressed by the authors.

Response: We thank you for your valuable comment and appreciate your positive evaluation of our manuscript.

REVIEWERS' COMMENTS

Reviewer #2 (Remarks to the Author):

Many thanks to the authors for their thoughtful response. This interpretation of hydrogen-bond linearity being promoted in stressed chains strikes me as very unlikely. Perhaps the all-trans arrangement being correlated with higher local dipoles and thus greater electrostatic interactions between chains is more likely. Instead, I think it is much more likely that "preferred" H-bond geometries are formed in conformationally relaxed states rather constrained conformational states. It is not clear to me that the prior literature cited by the authors supports the interpretation provided by the authors in the revised SI, vs simply conformational relaxations enabling a greater number of hydrogen bonding interactions as is not the case here.

Given the other reviewer comments, I will not stand in the way of publication, but I would urge that the authors think very carefully about how strongly they want to argue specific mechanism. The present explanation at best seems highly speculative to me. If they want to stick to better hydrogen bonding, though, they should at least revise the structures in the new SI figure. While the preferred NH - OC geometry is linear from N-H-O, it NOT linear H-O-C. Instead, an angle of 120 deg is preferred, as this involves the site of greatest electron density on the oxygen atom.

This is illustrative of my main concern -- the authors rationalize the effect in terms of structural alignment of a precise molecular orientation, but argue for the wrong molecular orientation! It makes clear that there is no actual reason a priori to invoke a certain preferred alignment. Put another way, there is no insight into how one would design a polymer to promote the desired increased linearity of interchain H-bonds.

Also, I would encourage that the authors look carefully at the energetic differences involved with slight perturbations of H-bond geometry. H-bonds are directional, but not quite as directional as many assume from the term. In practice, small differences in angle tend not to matter very much.

Point-to-point responses to referees

Contents

Responses to reviewer #2: Page 2

Reviewers' comments are in italic

Responses are highlighted in gray

The updated contents are highlighted in yellow

The dragged contents from the original manuscript are in sky-blue

The materials only for revision are in purple

[Response to Reviewer #2]

Comment: Many thanks to the authors for their thoughtful response. This interpretation of hydrogen-bond linearity being promoted in stressed chains strikes me as very unlikely. Perhaps the all-trans arrangement being correlated with higher local dipoles and thus greater electrostatic interactions between chains is more likely. Instead, I think it is much more likely that "preferred" H-bond geometries are formed in conformationally relaxed states rather than constrained conformational states. It is not clear to me that the prior literature cited by the authors supports the interpretation provided by the authors in the revised SI, vs simply conformational relaxations enabling a greater number of hydrogen bonding interactions as is not the case here.

Given the other reviewer comments, I will not stand in the way of publication, but I would urge that the authors think very carefully about how strongly they want to argue specific mechanism. The present explanation at best seems highly speculative to me. If they want to stick to better hydrogen bonding, though, they should at least revise the structures in the new SI figure. While the preferred NH - OC geometry is linear from N-H-O, it NOT linear H-O-C. Instead, an angle of 120 deg is preferred, as this involves the site of greatest electron density on the oxygen atom.

This is illustrative of my main concern -- the authors rationalize the effect in terms of structural alignment of a precise molecular orientation, but argue for the wrong molecular orientation! It makes clear that there is no actual reason a priori to invoke a certain preferred alignment. Put another way, there is no insight into how one would design a polymer to promote the desired increased linearity of interchain H-bonds.

Also, I would encourage that the authors look carefully at the energetic differences involved with slight perturbations of H-bond geometry. H-bonds are directional, but not quite as directional as many assume from the term. In practice, small differences in angle tend not to matter very much.

Response: We sincerely appreciate your valuable comment. To further clarify the explanation we have modified the manuscript. We hope that you find our responses sufficient and satisfying.

As the reviewer has suggested, we accept the possibility of different interpretations. We have added the sentences in "Discussion" section of the manuscript.

- (Lines 388-391), In the next research, it might be necessary to perform fundamental study at molecular level and also careful interpretation of H-bonding behavior in the self-healing TPU due to complex influences of other non-covalent bonds and polymer conformation.

In addition, as the reviewer pointed out about the bond angle of H-O-C, we have removed Figure S12b as follows to avoid confusion. Please find the new Figure S12.

Supplementary Figure 12. (a) Stretching-dependent change of H-bond states and related FT-IR bands of C-IP-SS with a constant number of H-bond donors (carbonyl groups) and (b) the effect of chain conformational change upon straining on the linearity and strength of H-bond.